# Efficient Hallucination Detection for LLMs Using Uncertainty-Aware Attention Heads

Artem Vazhentsev [1]  Lyudmila Rvanova [2]  Gleb Kuzmin [2]  Ekaterina Fadeeva [3]  Ivan Lazichny [4]
Alexander Panchenko [5,6]  Maxim Panov [1]  Mrinmaya Sachan [3]  Preslav Nakov [1]  Timothy Baldwin [1]
Artem Shelmanov [1]

## Abstract

While large language models (LLMs) have become highly capable, they remain prone to factual inaccuracies, commonly referred to as "hallucinations." Uncertainty quantification (UQ) offers a promising way to mitigate this issue, but most existing methods are computationally intensive and/or require supervision. In this work, we propose Recurrent Attention-based Uncertainty Quantification (RAUQ), an unsupervised and efficient framework for identifying hallucinations. The method leverages an observation about transformer attention behavior: when incorrect information is generated, certain "uncertainty-aware" attention heads tend to reduce their focus on preceding tokens. RAUQ automatically detects these attention heads and combines their activation patterns with token-level confidence measures in a recurrent scheme, producing a sequence-level uncertainty estimate in just a single forward pass. Through experiments on twelve datasets spanning question answering, summarization, and translation across nine different LLMs, we show that RAUQ consistently outperforms state-of-the-art UQ baselines. Importantly, it incurs minimal overhead, requiring less than 1% additional computation. Since it requires neither labeled data nor extensive parameter tuning, RAUQ serves as a lightweight, plug-and-play solution for real-time hallucination detection in white-box LLMs.[1]

## 1. Introduction

Large Language Models (LLMs) have become the de facto backbone of modern Natural Language Processing (NLP) systems; yet, the impressive fluency of their responses often conceals various inconsistencies known as "hallucinations" (Huang et al., 2025). There are several ways to address hallucinations, such as post-hoc verification using external knowledge bases (Min et al., 2023), incorporating retrieval-augmented generation to ground outputs in factual data (Lewis et al., 2020), or filtering/altering responses based on the uncertainty quantification (UQ) (Kuhn et al., 2023; Farquhar et al., 2024). The latter approach is the focus of this work.

Uncertainty is a fundamental concept in machine learning, reflecting the fact that we usually lack complete information about the model's predictions or parameters (Gal & Ghahramani, 2016; Houlsby et al., 2011; Hüllermeier & Waegeman, 2021; Tonolini et al., 2024). High predictive uncertainty typically signals a greater likelihood of hallucinations in the model output. Unlike verification methods that rely on external knowledge sources to detect hallucinations, UQ leverages the model's internal capabilities, thereby mitigating issues related to the completeness of external sources and offering greater versatility. As shown in previous work, uncertainty and confidence scores can be used to detect hallucinations that arise due to limitations of LLM's parametric knowledge or due to the ambiguity of the requests in various generation tasks (Malinin & Gales, 2021; Geng et al., 2024; Baan et al., 2023), including question-answering, machine translation, text summarization, and speech recognition.

Uncertainty quantification for classification and regression tasks is a well-established area spanning decades of research (Zhang et al., 2019; He et al., 2020; Xin et al., 2021; Wang et al., 2022; Vazhentsev et al., 2023; He et al., 2024b). At the same time, UQ for generative tasks has only recently emerged as an active topic and still presents open challenges. A crucial difference from classification is that an LLM performs not a single but multiple conditionally dependent predictions.

[1]Mohamed bin Zayed University of Artificial Intelligence (MBZUAI), Abu Dhabi, United Arab Emirates [2]FusionBrain Lab, Moscow, Russia [3]ETH Zurich, Zurich, Switzerland [4]Independent Researcher [5]Applied AI Institute, Moscow, Russia [6]Computational Semantics Lab, Moscow, Russia. Correspondence to: Artem Vazhentsev <Artem.Vazhentsev@mbzuai.ac.ae>, Artem Shelmanov <Artem.Shelmanov@mbzuai.ac.ae>.

*Proceedings of the $43^{rd}$ International Conference on Machine Learning*, Seoul, South Korea. PMLR 306, 2026. Copyright 2026 by the author(s).

[1]https://github.com/mbzuai-nlp/rauq-hallucination-detection

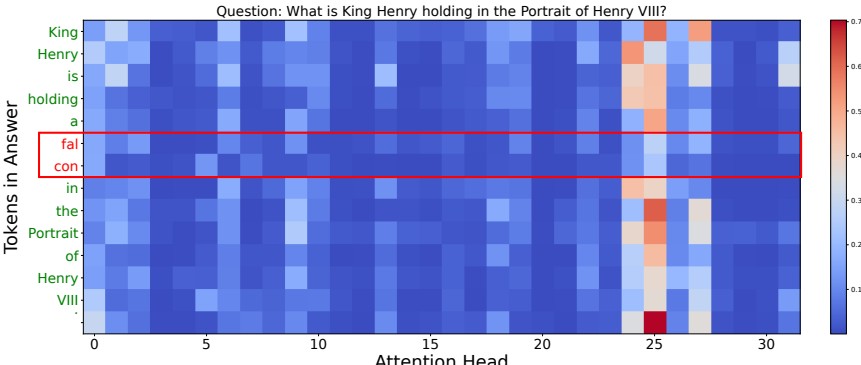

*Figure 1.* Attention weights in the 29th layer of Llama-3.1 8B from each generated token to its preceding token, given the prompt *What is King Henry holding in the Portrait of Henry VIII?*. The $y$ axis specifies the generated tokens, and the $x$ axis specifies the attention heads. Warmer colors indicate higher attention values. The output contains the factually incorrect token *falcon* (the correct answer is *gloves* and *dagger*). Notably, the 25th attention head stands out by consistently assigning relatively high attention to the preceding token. However, for the hallucinated token *falcon*, this attention drops sharply – potentially serving as a signal for hallucination detection.

While recent work has proposed several promising techniques for quantifying predictive uncertainty in generation, e.g. (Kuhn et al., 2023; Farquhar et al., 2024; Duan et al., 2024; Qiu & Miikkulainen, 2024; Lin et al., 2024b), existing approaches exhibit certain limitations in practice. Namely, information-based scores such as sequence probability (SP) and token-level entropy are simple and fast, but they often underperform on long-form generation tasks (Zhang et al., 2024; Vazhentsev et al., 2025a). Sampling-based scores offer stronger performance but incur large computational overhead (Kuhn et al., 2023; Lin et al., 2024b; Vashurin et al., 2025). Supervised confidence regressors (Azaria & Mitchell, 2023; CH-Wang et al., 2024), i.e., thin supplementary modules trained on supervised annotation, yield accurate scores but require costly, task-specific annotations and often fail to generalize to out-of-distribution data or across tasks (Vazhentsev et al., 2025a). Thus, despite the recent surge of developments in UQ for LLMs, there is still a lack of an effective, versatile UQ method that (*i*) avoids the high computational costs associated with sampling-based approaches, and (*ii*) is robust across tasks and domains.

In this work, we aim to construct such a method. For this purpose, we peek into the attention weights of the transformer and identify patterns that are highly indicative of the presence of hallucinations. Self-attention matrices encode how strongly each newly generated token attends to its immediate context. We empirically observe a systematic drop in the attention weight to the preceding tokens in specific attention heads precisely at positions where the model later proves to be factually incorrect (Figure 1). Based on this finding, we argue that a small number of attention heads capture the behavior of transformer-based LLMs under uncertainty. We propose a method that automatically identifies such "uncertainty-aware" heads inside individual LLM layers and extracts the token-level signal from them.

The method recurrently fuses this signal with token probabilities and confidence scores from previously generated tokens, thus capturing the conditional dependencies across generation steps. Finally, it aggregates the token-level scores across the generated sequence and layers. The resulting sequence-level uncertainty score achieves state-of-the-art performance and demonstrates high robustness to the choice of value for its single hyperparameter. Moreover, since the attention weights are readily available at inference time for white-box LLMs, the method requires no additional generation passes and adds almost no computational overhead to the response latency.

**Contributions:**

- **In-depth analysis** of attention-based patterns in LLMs associated with hallucinations, which uncovers what we term "uncertainty-aware" heads, i.e., attention heads whose signals notably correlate with hallucination occurrences.

- **RAUQ** (Recurrent Attention-based Uncertainty Quantification) – an *unsupervised* UQ method that turns raw attentions and LLM probabilities into reliable uncertainty scores while adding only <1% latency. RAUQ requires *no* task-specific labels or hyperparameter tuning for a particular LLM, making it a plug-and-play method for white-box LLMs.

- **Thorough experimental evaluation** on nine LLMs and 12 benchmarks, spanning summarization, translation, and question answering, showing that RAUQ achieves state-of-the-art results over 15 baselines. We also demonstrate the importance of each component within the method and illustrate that each one individually could improve other UQ methods.

## 2. Related Work

Several recent studies have proposed attention-based UQ methods for detecting hallucinations in LLM generations.

Zhang et al. (2023) use attention weights to propagate uncertainty across generation steps by capturing conditional dependencies, helping to mitigate overconfidence from prior hallucinations. However, attention plays a secondary role, with the method mainly relying on probability and entropy. Yuksekgonul et al. (2024) conduct a mechanistic analysis of attention patterns associated with LLM factual errors and attribute hallucinations to reduced attention to so-called "constrained" tokens – prompt elements that restrict the scope of the response. They propose a supervised method (SAT probe) based on this finding and report modest improvements over baseline approaches. In a similar vein, Contextualized Sequence Likelihood (Lin et al., 2024a) leverages attention to important tokens in the input to reweigh the contributions of token logits when computing weighted sequence likelihood. Lookback Lens (Chuang et al., 2024) is based on the hypothesis that hallucinations correlate with less attention paid to the input context. The method computes the ratio of cumulative attention weights assigned to tokens in the generated answer versus the prompt and trains a linear classifier on these features. Attention-based features are also used in Trainable Attention-Based Dependency (TAD) (Vazhentsev et al., 2025a) as a proxy for conditional dependency. It also introduces recurrence when computing uncertainty for subsequent tokens. TAD demonstrates strong results for in-domain tasks, outperforming Lookback Lens, but both methods lack generalization due to their supervised nature. Finally, Sriramanan et al. (2024) proposed the Attention Score method, where they compute a length-normalized sum of log attention weights to preceding tokens across the prompt and the answer as a confidence score.

Although recent studies show that attention weights offer valuable signals for detecting hallucinations in LLM outputs, existing methods suffer from various limitations that hinder their effectiveness. SAT Probe, Lookback Lens, and TAD are supervised and show limited generalization beyond their training domain. Zhang et al. (2023) and Lin et al. (2024a) leverage attention only as a supplement to other scores. Sriramanan et al. (2024) do not select proper attention heads before averaging, and allow the attention weights from prompt tokens to participate in the aggregation for the final score, which causes underperformance.

In this work, we aim to overcome the limitations of existing methods. To this end, we identify strong and generalizable attention-based patterns for LLM hallucination detection, isolate the key techniques required to effectively exploit these patterns, and develop a robust *unsupervised* uncertainty quantification method that achieves state-of-the-art performance.

## 3. Hallucination-Associated Patterns in Attention Maps

We analyze the model's attention maps when an LLM generates correct vs. incorrect outputs. We start with an analysis of attention weights to the immediately preceding token, i.e. $a_{i,i-1}^{l,h}$ – attention weight to the $(i-1)$-th token during the generation of the $i$-th token from the layer $l$ and attention head $h$. Let $N$ be the number of tokens in the answer, $H$ be the number of attention heads in each layer, and $L$ be the number of layers. For illustration, we use Llama-3.1 8B.

**Difference between attention weights for hallucinated and non-hallucinated tokens.** Figure 1 presents an example of the attention weights to preceding tokens $a_{i,i-1}^{l,h}$ in one of the LLM layers for the input question from the TruthfulQA dataset: *What is King Henry holding in the Portrait of Henry VIII?* Most of the generated tokens are aligned with the question. However, the token *falcon* represents a hallucination; the answer should be *gloves and dagger*.

For most attention heads, the weights to previous tokens remain low across all generated tokens. In contrast, the 25th head exhibits a distinct pattern: it assigns relatively high attention to the preceding token for non-hallucinated (i.e., correct) tokens, but this attention drops substantially for the hallucinated token *falcon*. This example demonstrates that attention weights from a small subset of attention heads are "uncertainty-aware", i.e., they are sensitive to generation factuality and could help identify hallucinations. More examples of the similar pattern for Llama and other LLMs are presented in Figures 5 to 8 in Appendix E. Moreover, similar patterns beyond QA are shown for Llama on summarization and translation in Figures 9 to 14 in Appendices E.2 and E.3. While the specific layer and head may vary, we consistently observe this pattern across all tested LLMs.

**Difference between average attention weights for incorrect and correct answers.** We select 10 correct and 10 incorrect answers generated by the LLM. To evaluate the correctness of each answer, we use AlignScore – a continuous metric that quantifies the semantic similarity between the generated response and the gold answer (Zha et al., 2023). We sort all generations by their AlignScore and designate the top 10 as correct answers and the bottom 10 as incorrect. Then, we compute the average attention weight to the previous token across all tokens in the answer using the attention heads in the 29th and 23rd layers of the LLM, i.e. $\bar{a}^{l,h} = \frac{1}{N-1} \sum_{i=2}^{N} a_{i,i-1}^{l,h}$. Figure 2 presents the resulting values, where each row corresponds to a single LLM answer and each column indicates the average attention weight from a specific head. These attention maps demonstrate that certain heads consistently assign higher average attention when the LLM generates correct answers as compared to incorrect ones. Moreover, there is a notable correlation between the quality of the answer and average attention (see Figure 3b).

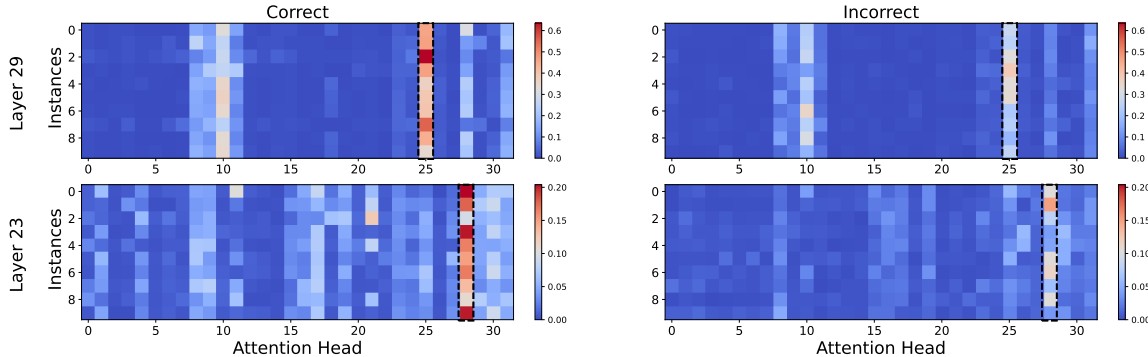

*Figure 2.* Average attention weights to the preceding token, aggregated over all answer tokens for questions from the TruthfulQA dataset using Llama-3.1 8B. The top 10 highest- and lowest-quality answers, as determined by a quality metric, are labeled as correct and incorrect, respectively. The black dashed box highlights the head with the highest average attention.

**Identifying uncertainty-aware heads.** Not all attention heads are useful for hallucination detection. To illustrate this, we compute the average attention score $\bar{a}^{l,h}$ across tokens in two scenarios: (1) attention values are averaged across all heads in a layer, i.e. $\bar{a}^l = \frac{1}{H} \sum_{h=1}^{H} \bar{a}^{l,h}$; (2) attention values are extracted from a single selected head with the *highest* average attention across tokens, i.e. $\bar{a}^{l,\mathbf{h}_l}$, where $\mathbf{h}_l = \arg\max_{h=1...H} \bar{a}^{l,h}$. Figure 3a compares the resulting values for correct and incorrect answers, showing both absolute and percentage differences. When relying only on the selected attention head, we observe a clear difference between correct and incorrect answers. In contrast, averaging attention across all heads largely obscures this signal. This once again highlights the importance of focusing on specific *uncertainty-aware* heads, which, as demonstrated in our analysis, can be identified by their consistently high average attention weights across tokens.

**Do we need to look further back at preceding tokens to better detect hallucinations?** We analyze the attention weights to multiple preceding tokens. Here, we compute $a_{i,i-k}^{l,h}$ – an attention weight to the $(i-k)$-th token ($k$-th preceding token), $k = 1, \ldots, 6$. Figure 3c shows the absolute differences and relative differences in percent between the average attention weights of the correct and incorrect answers. The attention weights differ substantially between correct and incorrect answers only for the two preceding tokens, with almost zero differences observed for earlier tokens. Notably, the difference is substantially larger for the first preceding token as compared to the second one.

**Summary.** Our analysis uncovers attention patterns associated with the factuality of individual tokens and LLM responses in general. A key observation is that such patterns emerge only for a small subset of specific attention heads. Effectively leveraging them requires first identifying the relevant uncertainty-aware attention heads. We also observe that the immediately preceding token provides the strongest signal, and thus we focus solely on it in our method design.

# 4. RAUQ: Recurrent Attention-Based Uncertainty Quantification Method

**Key ideas and theoretical grounding.** RAUQ integrates three key ideas:

(1) The first idea is that *attention weights to previous tokens contain patterns indicative of the presence of hallucination.* These patterns are a natural consequence of LLM pre-training, where the objective is to maximize the likelihood of given texts. Mechanistically, attention to the previous token plays a special role: it supports local syntactic and semantic continuity, stabilizes incremental belief propagation, and reflects the model's confidence that the current token follows naturally from the immediately preceding one. When hallucinating, the model relies more on global priors about what typically comes next or on coarse topic-level representations rather than on token-level grounding, causing attention to shift away from the immediate context.

As discussed in Section 2, few works have explored and motivated attention as a signal for hallucination detection (Zhang et al., 2023; Yuksekgonul et al., 2024; Lin et al., 2024a; Chuang et al., 2024; Vazhentsev et al., 2025a; Sriramanan et al., 2024). The most principled approach is proposed by Sriramanan et al. (2024), who show that attention weights exhibit systematic patterns indicative of hallucinations, revealed through eigenvalue analysis of attention kernels. They use only the attention weights to the previous token, as these correspond to the eigenvalues of the lower triangular attention matrix, and their sum exactly equals its log determinant. We reveal a similar pattern through an empirical mechanistic analysis of attention weights, examining the correlation between hallucinations and attention weight distributions, as shown in Section 3.

(2) The second idea is the idea of *attention head selection*. We observe that the majority of heads are not indicative of hallucinations (as shown in Figure 3a).

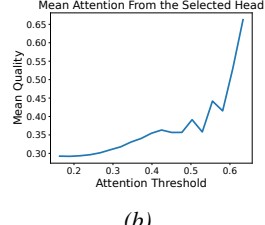
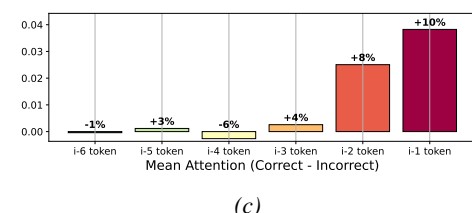

*Figure 3.* Comparison between incorrect (AlignScore < 0.1) and correct (AlignScore > 0.9) answers in average attention weights to preceding tokens during answer generation for questions from the TruthfulQA dataset using Llama-3.1 8B. (a) Attention values to the last preceding token only are presented for two scenarios: (left) from the selected head with the highest average attention; (right) averaged across all heads. (b) The relationship between average response quality and the average attention weight to the last preceding token only in the selected head. (c) Difference between correct and incorrect answers in average attention weights to more preceding tokens.

Therefore, we suggest selecting the most contrastive head that has the best potential for discriminating between hallucinations and non-hallucinations. Our findings are well supported by prior mechanistic interpretability studies of attention heads, which have shown that different heads serve distinct functions (Elhelo & Geva, 2025).

(3) In the third idea that RAUQ integrates, we follow (Zhang et al., 2023; Vazhentsev et al., 2025a) and acknowledge that computing the confidence at generation step $i$ requires propagating the uncertainty from the previous steps due to the conditional dependencies in the probability distribution modeled by the LLM. Namely, even if the previous tokens were generated with low confidence, a model may condition on them and be highly confident in its erroneous current token prediction. In order to address this issue, we introduce a formulation that *recurrently propagates the confidence from the previous steps*. Here, attention also plays a key role, as it is widely used as a proxy for the dependence of the current prediction on the preceding generations (Zhang et al., 2023).

Guided by these ideas and robust design choices, we develop a principled and reliable method for hallucination detection (see Algorithm 1).

**Selecting an informative attention head in each layer.** Let $\mathbf{x} = (x_1 x_2 \ldots x_M)$ be the input sequence, and let $\mathbf{y} = (y_1 y_2 \ldots y_N)$ denote the corresponding generation of length $N$ from a LLM with $L$ layers and $H$ attention heads. Let $a_{i,i-1}^{l,h}$ be the attention weight from token $y_i$ to $y_{i-1}$ computed by the $h$-th head in layer $l$. For each layer $l$, we select the head with the maximum average attention weights between consecutive tokens:

$$\mathbf{h}_l(\mathbf{y}) = \arg\max_{h=1\ldots H} \frac{1}{N-1} \sum_{i=2}^{N} a_{i,i-1}^{l,h}. \quad (1)$$

By taking the maximum within each layer, the method effectively selects the most contrastive head with the best potential for discriminating between hallucinations and non-hallucinations.

---

**Algorithm 1** RAUQ: Recurrent Attention-based Uncertainty Quantification method, with $s_i = P(y_i \mid y_{<i}, \mathbf{x})$

---

1: **Data:** prompt $\mathbf{x}$, LLM generation $\mathbf{y} = y_{1:N}$, attention weights $a_{i,i-1}^{l,h}$ for each layer $l$ and head $h$, probabilities $P(y_i \mid y_{<i}, \mathbf{x})$, hyperparameter $\alpha$.
2: **Output:** Uncertainty score $\mathbf{u}(\mathbf{y})$
3: // Selection of uncertainty-aware heads
4: **for** $l = 1$ **to** $L$ **do**
5: $\quad \mathbf{h}_l \leftarrow \arg\max_{h=1\ldots H} \frac{1}{N-1} \sum_{i=2}^{N} a_{i,i-1}^{l,h}$
6: **end for**
7: // Computing token-level confidence scores with uncertainty-aware heads
8: **for** $i = 1$ **to** $N$ **do**
9: $\quad$ **if** $i = 1$ **then**
10: $\quad\quad \mathbf{c}_l(y_i) \leftarrow P(y_i \mid \mathbf{x})$
11: $\quad$ **else**
12: $\quad\quad \mathbf{c}_l(y_i) \leftarrow \alpha P(y_i \mid y_{<i}, \mathbf{x}) + (1-\alpha) a_{i,i-1}^{l,\mathbf{h}_l} \mathbf{c}_l(y_{i-1})$
13: $\quad$ **end if**
14: **end for**
15: // Computing layer-wise and final uncertainty scores
16: $\mathbf{u}_l(\mathbf{y}) \leftarrow -\frac{1}{N} \sum_{i=1}^{N} \log \mathbf{c}_l(y_i)$
17: $\mathbf{u}(\mathbf{y}) \leftarrow \max_{l \in \mathcal{L}} \mathbf{u}_l(\mathbf{y})$
18: **return** $\mathbf{u}(\mathbf{y})$

---

**Token-level layer-wise recurrent confidence score.** To propagate uncertainty from previous generation steps on which the current token most strongly depends, we recurrently compute the confidence score $\mathbf{c}_l(y_i)$ for the $i$-th token by leveraging the confidence of the previous token $\mathbf{c}_l(y_{i-1})$, the attention weight $a_{i,i-1}^{l,\mathbf{h}_l}$ from the selected head $\mathbf{h}_l = \mathbf{h}_l(\mathbf{y})$, and a token-level confidence signal $s_i$, which is a conditional probability of the current token $s_i = P(y_i \mid y_{<i}, \mathbf{x})$ by default:

$$\mathbf{c}_l(y_i) = \begin{cases} s_1, & \text{if } i = 1, \\ \alpha s_i + (1-\alpha) a_{i,i-1}^{l,\mathbf{h}_l} \mathbf{c}_l(y_{i-1}), & \text{if } i > 1, \end{cases} \quad (2)$$

where $\alpha$ is a hyperparameter that balances the contributions of each component. This recurrent formulation also helps to avoid an explosion in confidence scores with an increase in sequence length. We present an ablation study on the impact of varying the parameter $\alpha$ in Section 5.3 and show that a single value of $\alpha$ provides robust performance across various tasks and even models.

It is possible to use various token-level confidence scores $s_i$. We additionally consider an entropy-based signal $s_i = \mathbf{c}_i^{\mathrm{ent}} = \log|\mathcal{V}| + \sum_{v\in\mathcal{V}} P(v \mid y_{<i}, \mathbf{x}) \log P(v \mid y_{<i}, \mathbf{x})$, where $\mathcal{V}$ is the vocabulary of the language model. We compare the probability- and entropy-based signals in Section 5.3. Based on this comparison, we use the probability-based signal for base models and the entropy-based signal for instruction-tuned models.

**Sequence-level layer-wise uncertainty score.** Sequence-level errors are typically either (1) *distributed* across all tokens, e.g. in the summarization task, or (2) *localized* in a single fact-related token, e.g. in the QA task. To take into account both cases in the sequence-level uncertainty score, similar to perplexity, we compute the mean logarithm of the confidence scores across all tokens in the reply (importantly, we do not aggregate the scores for the tokens in the prompt):

$$\mathbf{u}_l(\mathbf{y}) = -\frac{1}{N} \sum_{i=1}^{N} \log \mathbf{c}_l(y_i). \tag{3}$$

**Final uncertainty score.** Finally, we aggregate the layer-wise uncertainty scores in an unsupervised manner by taking the maximum score across layers, which provides an upper-bound estimate of uncertainty:

$$\mathbf{u}(\mathbf{y}) = \max_{l\in\mathcal{L}} \mathbf{u}_l(\mathbf{y}), \tag{4}$$

where $\mathcal{L}$ is a subset of layers. Following previous work (Azaria & Mitchell, 2023; Vazhentsev et al., 2025a), we use the intermediate layers of the model, as they are the most informative for hallucination detection. An ablation study with various aggregation functions is presented in Section 5.3.

# 5. Experiments

## 5.1. Experimental Setup

We conducted extensive experiments across three key generation tasks: question answering (QA), text summarization (Summ), and machine translation (MT). We evaluated the effectiveness of UQ in filtering unreliable outputs through selective generation.

For all base LLMs and tasks, we set $\alpha = 0.2$; for instruction-tuned models, we set $\alpha = 0.9$. In all cases, we use the same range of layers $\mathcal{L}$ – from the first third to the second third of the model (e.g., layers 10 to 22 for Llama-3.1 8B) without any tuning.

**Datasets.** We consider seven datasets for QA, three for Summ, and two for MT. Detailed dataset descriptions are provided in Appendix A.2, with statistics in Table 4.

**Models.** To show the generalization of the method across various models, we use several widely used open-weight LLMs: Llama-3.1 8B (Dubey et al., 2024), Qwen-2.5 7B (Yang et al., 2024), Gemma-2 9B (Rivière et al., 2024), and Falcon-3 10B (Falcon-LLM Team, 2024). Additionally, in Appendix C.1, we experiment with open-weight LLMs of varying sizes: SmolLM-2 360M (Allal et al., 2025), Llama-3.2 1B, and Llama-3.1 70B (Dubey et al., 2024). In Appendix C.2, we also experiment with open-weight instruction-tuned LLMs: Llama-3.1 8B Instruct (Dubey et al., 2024) and GPT-OSS 20B (OpenAI, 2025). Detailed descriptions of the generation parameters are presented in Table 4 in Appendix A.2. Different LLMs expose attention weights in various formats, so we apply a model-specific normalization that transforms raw attention outputs into a unified lower-triangular attention matrix.

**UQ baselines.** We compare RAUQ against 15 diverse UQ baselines, including both unsupervised and supervised methods. Detailed descriptions of all baselines are provided in Appendix A.1 and Appendix C.4.

**Evaluation measures.** As the main evaluation measure, we use the Prediction Rejection Ratio (PRR) (Malinin & Gales, 2021; Vashurin et al., 2025). PRR measures the area under the rejection curve, which plots the average quality of the remaining responses when we abstain from a fraction of the most uncertain predictions. We compute PRR over only the first 50% of the curve, as rejecting more than half of the instances is typically impractical. We evaluate PRR using task-specific quality measures: accuracy for MMLU and GSM8k; COMET (Rei et al., 2020) for MT; and Align-Score (Zha et al., 2023) for the rest. A detailed description of the PRR is provided in Appendix A.3. Additionally, we calculate ROC-AUC using discrete quality measures obtained by thresholding the original continuous values.

## 5.2. Main Results

Table 1 presents the mean PRR for each task (QA, Summ, and MT) for each evaluated LLM. To compute the mean PRR for each task, we average the PRR scores across all relevant datasets, for example, XSum, CNN, and SamSum for summarization. The aggregated PRR scores provide a robust measure of the performance of various methods for each task and model. Detailed results for each model and dataset are presented in Tables 21 to 24 in Appendix D. The ROC-AUC results are shown in Table 17 in Appendix C.3.

The results demonstrate that RAUQ consistently outperforms the previous state-of-the-art for the QA and translation tasks by a sizable margin across all evaluated LLMs.

*Table 1.* Mean PRR↑ across tasks for the evaluated LLMs. Warmer color indicates better results.

| UQ Method | Llama-3.1 8B | | | Qwen-2.5 7B | | | Gemma-2 9B | | | Falcon-3 10B | | | Mean |
|---|---|---|---|---|---|---|---|---|---|---|---|---|---|
| | QA | Summ | MT | QA | Summ | MT | QA | Summ | MT | QA | Summ | MT | |
| MSP | .347 | .296 | .397 | .329 | .151 | .369 | .361 | .334 | .381 | .345 | .177 | .333 | .318 |
| Perplexity | .347 | .419 | .380 | .343 | **.254** | .406 | .383 | .375 | .405 | .356 | .180 | .439 | .357 |
| CCP | .285 | .307 | .340 | .271 | .186 | .327 | .329 | .345 | .320 | .299 | .128 | .287 | .285 |
| Attention Score | .014 | .126 | .178 | .038 | .130 | .142 | .064 | .103 | .146 | .054 | **.192** | .089 | .106 |
| Focus | .320 | .335 | .361 | .264 | .186 | .380 | .416 | .340 | .385 | .313 | .139 | .362 | .317 |
| Simple Focus | .342 | .306 | .415 | .342 | .136 | .399 | .396 | .322 | .422 | .351 | .095 | .385 | .326 |
| DegMat NLI Score entail. | .306 | .118 | .239 | .356 | .154 | .275 | .337 | .138 | .259 | .352 | .132 | .222 | .241 |
| Ecc. NLI Score entail. | .274 | -.008 | .284 | .322 | .002 | .306 | .298 | .020 | .290 | .327 | .038 | .281 | .203 |
| EVL NLI Score entail. | .293 | .114 | .217 | .349 | .154 | .245 | .332 | .133 | .252 | .351 | .135 | .206 | .232 |
| Lexical Similarity Rouge-L | .250 | .131 | .324 | .334 | .131 | .327 | .306 | .161 | .342 | .285 | .084 | .275 | .246 |
| EigenScore | .232 | .078 | .285 | .298 | .061 | .302 | .267 | .106 | .226 | .247 | .051 | .236 | .199 |
| LUQ | .287 | .173 | .214 | .351 | .196 | .213 | .344 | .206 | .259 | .335 | .121 | .196 | .241 |
| Semantic Entropy | .254 | .117 | .315 | .281 | .092 | .317 | .291 | .126 | .337 | .320 | .133 | .291 | .240 |
| SAR | .310 | .170 | .370 | .351 | .153 | .393 | .361 | .235 | .414 | .334 | .094 | .337 | .294 |
| Semantic Density | .330 | .153 | .264 | .352 | .110 | .291 | .375 | .167 | .255 | .358 | .141 | .280 | .256 |
| RAUQ | **.396** | **.428** | **.452** | **.358** | .213 | **.438** | **.421** | **.392** | **.473** | **.392** | .181 | **.465** | **.384** |

For instance, for the translation task using Gemma-2 9B, RAUQ largely outperforms the second-best method, Simple Focus, by 0.051 of PRR. In contrast, other single-generation methods based on the attention weights, such as Focus and Attention Score, perform substantially worse.

For summarization, RAUQ also achieves the best results across most models, often with a notable margin over the second-best method. Notably, RAUQ improves upon the second-best method (Perplexity) for Gemma-2 9B by 0.017 in terms of PRR. The only exceptions are Qwen-2.5 7B and Falcon-3 10B, where Perplexity and Attention Score achieve the best performance, respectively. Nevertheless, RAUQ is the second best and consistently outperforms all other sampling-based baselines on average.

Overall, while methods such as MSP, Focus, or SAR might achieve top performance in specific settings, RAUQ demonstrates the most robust performance across all tasks and models, consistently ranking as the best or second-best method by average performance in a task.

Table 15 in Appendix C.1 provides experimental results with ≤1B and 70B LLMs, which demonstrate that RAUQ is the best method on average across a wide range of model sizes and tasks, highlighting its strong generalization ability.

Table 16 in Appendix C.2 presents results for instruction-tuned LLMs. They show that RAUQ is the best-performing method on average across a range of model architectures, confirming that the attention pattern generalizes to instruction-tuned and mixture-of-experts (MoE) models.

Tables 18 and 19 in Appendix C.4 show comparison with supervised UQ methods. While RAUQ slightly underperforms supervised methods on their in-domain data, it greatly outperforms them on average in out-of-domain scenarios.

## 5.3. Hyperparameter Sensitivity and Ablation Studies

**Impact of the hyperparameter $\alpha$.** The hyperparameter $\alpha$ in Equation (2) balances the contributions of attention, confidence from the previous token, and the token-level confidence $s_i$. If $s_i$ is defined as the conditional probability of the current token, then setting $\alpha = 1$ makes RAUQ equivalent to perplexity. When $\alpha$ approaches 0, RAUQ relies solely on the attention weights from the selected head. Figure 4 presents the impact of $\alpha$ on the performance of the RAUQ method for Llama-3.1 8B. For all tasks, except MMLU, the best possible performance is achieved with $\alpha$ between 0.2 and 0.5.

While dataset-specific fine-tuning of $\alpha$ can lead to further improvements, we do not perform such careful tuning in our experiments (Table 1). Instead, we select $\alpha$ using a small out-of-domain subset for Llama-3.1 8B and apply this value uniformly *across all datasets and base LLMs*. Despite this, RAUQ achieves consistently strong performance across tasks and LLMs, often achieving top or near-top results. Strong performance with a fixed hyperparameter underscores the method's robustness.

**Aggregation functions.** Table 5 in Appendix B.1 compares the performance of our RAUQ method using various aggregation functions of token-level confidence scores. We experiment with four aggregation strategies: mean, median, sum of logarithms (inspired by MSP), and mean of logarithms (inspired by perplexity). For the summarization tasks and certain QA datasets such as SciQ, TriviaQA, and GSM8k, mean aggregation yielded the best performance. For MMLU, the sum of logarithms substantially outperformed other aggregation strategies, while median aggregation performed second-best for the MedQUAD and the TruthfulQA datasets.

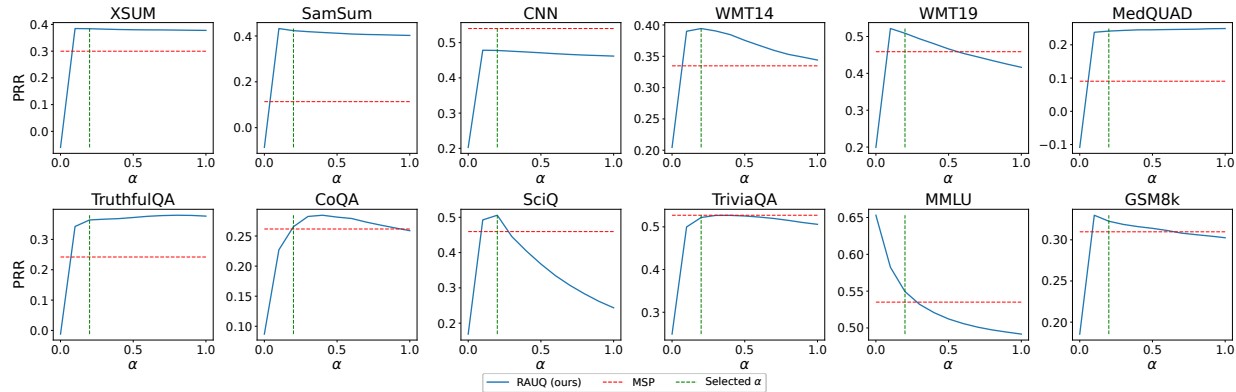

*Figure 4.* PRR↑ as a function of the hyperparameter $\alpha$ for Llama-3.1 8B. The vertical line marks the value of $\alpha$ used in our experiments.

Overall, however, the top two best performing methods were those that apply length normalization. Among them, the mean of logarithms of token-level confidence scores used in RAUQ consistently yielded the strongest results across all datasets.

Table 6 in Appendix B.1 compares RAUQ performance using various aggregation functions of layer-wise uncertainty scores. We consider three aggregation strategies: mean, median, and maximum. Both maximum and median yield similarly strong performances, while the mean aggregation performs slightly worse. Given that the maximum is a more intuitive choice, it effectively captures the peak uncertainty within a generation. It achieves better results in 6 out of 12 datasets, with a slight average improvement of 0.001 PRR across tasks over the median. Therefore, we adopt it as the default layer-wise aggregation method in our experiments.

**Recurrent uncertainty propagation functions.** Table 7 in Appendix B.1 presents the performance of the RAUQ method using various recurrent formulas for the calculation of token-level confidence scores. We consider five modifications of Equation (2): (1) removing attention weights, (2) removing recurrence, (3) replacing the confidence score of the previous token with its probability, (4) multiplying probabilities by attention weights, and (5) the recurrent formula proposed in RAUQ.

The proposed formula achieves the best results on the majority of the datasets. Removing either recurrence or attention often leads to substantially worse performance. The results highlight the importance of each component in the proposed formula for achieving good results.

**Attention change-point score.** We further evaluate variants of the attention term that detect hallucinations as a relative drop, or a change point, from the typical attention weight of the selected head. Table 8 in Appendix B.1 presents the performance of RAUQ with various variants of the attention term.

We evaluate three variants of the attention component in Equation (2): (1) subtracting a rolling average of the previous five attention scores, (2) subtracting the running average of all preceding attention scores, and (3) replacing the raw attention score with its ratio to the running average of all preceding scores. These variants improve performance on some individual datasets, but they do not consistently outperform the original formulation on average. This suggests that the absolute attention weight assigned to the immediately preceding token by the selected uncertainty-aware head is already a strong and robust indicator of hallucination.

**Token-level confidence signal.** Tables 9 and 10 in Appendix B.1 compare probability-based and entropy-based token-level confidence signals (see Equation (2)) for the Llama-3.1 8B base and instruction-tuned models. The two signals perform similarly for the base model, whereas the entropy-based signal consistently yields better results for the instruction-tuned model. These results indicate that entropy-based confidence is advantageous primarily in the instruction-tuned setting.

**Layers and heads selection.** Table 11 in Appendix B.2 shows RAUQ's performance across various layer subsets. The results indicate that using a subset of middle layers consistently achieves strong results, while selecting an optimal single layer offers only minor improvements and requires supervision. Tables 12 and 13 in Appendix B.2 show selected attention heads for WMT14 De-En and CoQA. The most informative heads are highly consistent within each task and largely overlap across tasks, supporting our finding that certain attention heads are particularly sensitive to hallucinations. Table 14 in Appendix B.2 shows results when a single optimal head per layer is selected on a small validation set. Average performance across all datasets remains similar, indicating that our dynamic, fully unsupervised strategy for selecting uncertainty-aware heads achieves near-optimal performance without task-specific tuning, thereby preserving RAUQ's plug-and-play applicability.

*Table 2.* PRR↑ for Llama-3.1 8B across various modifications of the Attention Score method incorporating components from RAUQ. The best method is in **bold**, the second best is underlined.

| UQ Method | XSum | SamSum | CNN | WMT14 | WMT19 | MedQUAD | TruthfulQA | CoQA | SciQ | TriviaQA | MMLU | GSM8k | Mean |
|---|---|---|---|---|---|---|---|---|---|---|---|---|---|
| Attention Score | .036 | .083 | .258 | .176 | .179 | -.295 | .081 | -.028 | -.142 | .067 | .209 | .209 | .069 |
| Attention Score (Gen. Tokens) | .020 | .117 | .261 | .196 | .198 | -.305 | -.020 | .064 | .124 | .130 | .232 | .192 | .101 |
| Attention Score (Gen. Tokens, Selected Head) | .154 | -.043 | .351 | .187 | .200 | -.113 | -.025 | .092 | .161 | .151 | .414 | .197 | .144 |
| RAUQ | **.370** | **.464** | **.452** | **.394** | **.509** | **.241** | **.364** | **.265** | **.506** | **.522** | **.549** | **.323** | **.413** |

*Table 3.* Inference runtime of UQ methods measured on all test instances from all datasets with generations from Llama-3.1 8B. The best results are in **bold**.

| UQ Method | Runtime per instance (s) | Overhead |
|---|---|---|
| MSP | $1.16_{\pm 0.45}$ | - |
| DegMat NLI Score Entail. | $6.40_{\pm 1.76}$ | 450% |
| Lexical Similarity ROUGE-L | $6.11_{\pm 1.75}$ | 425% |
| Semantic Entropy | $6.40_{\pm 1.76}$ | 450% |
| SAR | $10.71_{\pm 3.21}$ | 820% |
| Semantic Density | $6.27_{\pm 1.76}$ | 438% |
| RAUQ | $1.17_{\pm 0.45}$ | **0.3%** |

**Replacing attention with an interpretability score.** Table 20 in Appendix C.5 reports RAUQ performance when Layer Integrated Gradients (LIG) (Sundararajan et al., 2017) are used instead of attention scores. Specifically, we replace the original attention weights with LIG scores computed for the output projection layer and partition these scores to match the original multi-head structure. The results show only a 0.4% average drop in PRR, indicating that RAUQ does not critically rely on standard attention weights and can potentially be extended to models with non-standard attention mechanisms or architectures without attention.

**Extending our findings to the Attention Score method.** To demonstrate the robustness and generalization of the RAUQ components, we integrated them into the recently published Attention Score (AS) method (Sriramanan et al., 2024), resulting in two modifications. We compare (1) the original official implementation of AS, (2) AS that uses only the attention weights of the generated tokens, excluding the prompt, (3) AS that combines the previous modification and also implements the selection of the uncertainty-aware attention heads proposed here, and (4) the full RAUQ method with recurrence. The results in Table 2 show that excluding the contributions from the prompt tokens substantially improves the Attention Score, yielding a 0.032 improvement in PRR. The highest improvement is achieved on SciQ, CoQA, and TriviaQA. Incorporating the proposed attention head selection further boosts the average performance by 0.043, with a particularly large gain of 0.182 on MMLU. Finally, the full RAUQ, which additionally incorporates token probabilities and recurrently aggregates uncertainty scores from previous generation steps, yields a further advantage. These results suggest that our findings on attention heads and design choices in RAUQ generalize to prior UQ methods.

**Qualitative analysis.** We analyze samples with the highest and lowest RAUQ scores for Llama-3.1 8B on TruthfulQA. RAUQ effectively detects erroneous generations, with most of the detected errors attributed to factual and reasoning errors. Most of the erroneous generations with low uncertainty correspond to common misbeliefs. The detailed results are presented in Table 25 in Appendix F.

### 5.4. Computational Efficiency

To demonstrate the computational efficiency of RAUQ, we conducted an extensive runtime comparison against other state-of-the-art UQ methods using Llama-3.1 8B. All experiments use single-instance batches on a single 80GB NVIDIA H100 GPU, following the same setup as in Section 5.2. Table 3 reports the average runtime per instance for each UQ method and quantifies their computational overhead relative to standard LLM inference without UQ.

State-of-the-art UQ methods such as DegMat (Lin et al., 2024b), Semantic Entropy (Kuhn et al., 2023), and SAR (Duan et al., 2024) introduce huge computational overhead (400-800% of the original inference time) due to repeated sampling. In contrast, RAUQ introduces less than 1% overhead since it does not require sampling or inference of an auxiliary model, making it a fast, lightweight, and plug-and-play solution for any white-box LLM.

### 6. Conclusion and Future Work

We introduced RAUQ, an unsupervised, attention-based framework that converts the intrinsic signals already produced by every transformer layer into reliable sequence-level uncertainty scores with a single forward pass. A simple head-selection heuristic, a recurrent confidence propagation rule, and a length-normalized aggregation allow RAUQ to capture LLM confidence without external supervision or multiple sampling. Extensive experiments on twelve datasets spanning question answering, abstractive summarization, and machine translation, and on nine open-weight LLMs show that RAUQ delivers state-of-the-art performance. Moreover, RAUQ adds only <1 % latency overhead, making it a practical off-the-shelf UQ technique.

In future work, we plan to extend RAUQ to black-box LLMs, to improve calibration in deployment settings, and to broaden its applicability to task-specific diagnostics and diverse generation error types beyond hallucinations.

## Acknowledgments

We would like to thank the anonymous reviewers for their constructive comments, which have helped us improve the quality of this paper.

## Impact Statement

In this work, we proposed RAUQ, a plug-and-play method for real-time hallucination detection in white-box Large Language Models (LLMs), which requires no task-specific labels or multiple samples. RAUQ is efficient, easy to integrate, and demonstrates substantial performance improvements over baseline methods in our experiments. We believe that our work is a meaningful step toward more trustworthy and responsible use of LLMs, particularly in safety-critical domains such as healthcare and legal documentation. In our experiments, we considered open-weight LLMs and datasets not aimed at harmful content. We do not identify direct negative social impacts from our approach, as it does not rely on sensitive data, user annotations, or other elements that might raise ethical concerns. However, RAUQ should be viewed as a complement to, rather than a replacement for, external verification and human oversight, particularly in safety-critical settings.

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

# A. Additional Details of Experimental Setup

## A.1. Detailed Description of UQ Baselines

We compare the proposed RAUQ method to 16 diverse UQ baselines (15 core). As a sanity check, we include simple unsupervised baselines such as Maximum Sequence Probability (MSP), Mean Token Entropy (MTE), and Perplexity (Fomicheva et al., 2020). Among state-of-the-art baselines for white-box LLMs, we compare our method to Semantic Entropy (Kuhn et al., 2023), hallucination detection with a stronger focus ("Focus") (Zhang et al., 2023), Claim-Conditioned Probability ("CCP") (Fadeeva et al., 2024), EigenScore (Chen et al., 2024), Shifting Attention to Relevance ("SAR") (Duan et al., 2024), Semantic Density (Qiu & Miikkulainen, 2024), and Attention Score (Sriramanan et al., 2024). We also experiment with the "Simple Focus" method, which is a simplified variant of the "Focus" method (Zhang et al., 2023). It preserves only the core scoring components: attention-based signals and greedy log-likelihood, while omitting a proxy model, IDF-based keywords, and NER. Additionally, we consider UQ methods for black-box LLMs, as they also demonstrate strong performance in recent works despite not having access to logits or their hidden states. We use Lexical Similarity based on Rouge-L (Fomicheva et al., 2020), Long-text Uncertainty Quantification ("LUQ") (Zhang et al., 2024), and methods from (Lin et al., 2024b) – Degree Matrix ("DegMat"), Eccentricity, and Sum of Eigenvalues of the graph Laplacian ("EVL").

## A.2. Dataset and Generation Statistics

For QA, we use seven datasets: TruthfulQA (Lin et al., 2022) – a benchmark for assessing the truthfulness of LLM responses, SciQ (Welbl et al., 2017) for scientific QA, MMLU (Hendrycks et al., 2021) – a standard multitask evaluation benchmark, TriviaQA (Joshi et al., 2017) for trivia questions, CoQA (Reddy et al., 2019) for conversational QA, MedQUAD (Ben Abacha & Demner-Fushman, 2019) for medical questions, and GSM8k (Cobbe et al., 2021) for mathematical reasoning. For summarization, we used three datasets with different summarization types: CNN/DailyMail (See et al., 2017) for news article summarization, SamSum (Gliwa et al., 2019) for dialogue summarization, and XSum (Narayan et al., 2018) for summarizing into a single sentence. For the MT task, we evaluate on two pairs of languages from WMT: German–English from WMT19 (Barrault et al., 2019) and French–English from WMT14 (Bojar et al., 2014).

The detailed description of the used datasets and the generation parameters of LLMs is presented in Table 4. For all LLMs, we used the same generation hyperparameters, while for each dataset, we separately fixed the number of few-shot and maximum generation length. We use greedy decoding to generate the main sequence, for which we compute uncertainty, while sampling is used solely to obtain multiple outputs for sampling-based baselines. Consequently, the MSP score is always computed on the greedy output sequence (Aichberger et al., 2026).

*Table 4.* Statistics of the datasets and generation parameters of the LLMs. For all datasets, we do not limit the maximum input length.

| Task | Dataset | Number of test samples | N-shot | Generation length | Do sample | Temperature | Top-p | Beams | Repetition Penalty |
|------|---------|------------------------|--------|-------------------|-----------|-------------|-------|-------|--------------------|
| QA | TruthfulQA | 817 | 5 | 128 | | | | | |
| | SciQ | 1000 | 0 | 20 | | | | | |
| | MMLU | 2000 | 5 | 3 | | | | | |
| | TriviaQA | 2000 | 5 | 20 | | | | | |
| | CoQA | 2000 | all preceding questions | 20 | False | 1.0 | 1.0 | 1 | 1 |
| | MedQUAD | 1000 | 5 | 128 | | | | | |
| | GSM8k | 1319 | 5 | 256 | | | | | |
| Summ | CNN/DailyMail | 2000 | 0 | 128 | | | | | |
| | SamSum | 819 | 0 | 128 | | | | | |
| | XSum | 2000 | 0 | 128 | | | | | |
| MT | WMT19 (DE-EN) | 2000 | 0 | 107 | | | | | |
| | WMT14 (FR-EN) | 2000 | 0 | 107 | | | | | |

## A.3. Detailed Description of PRR

Prediction Rejection Ratio (PRR) (Malinin & Gales, 2021; Vashurin et al., 2025) is a measure that is upper-bounded by one computed as the area under the rejection curve, which plots the average quality of the retained responses when we abstain from a fraction of the most uncertain predictions. Following prior work (Vashurin et al., 2025), we computed PRR only over the first 50% of the rejection curve, as rejecting more than half of the instances is typically impractical in real-world applications.

The metric is normalized so that a PRR of zero or below indicates performance at or below the level of random chance correspondingly, while values approaching one reflect optimal performance. PRR is analogous to ROC-AUC or PR-AUC; however, unlike them, it can be applied not only to discrete quality measures (e.g., correct vs. incorrect answers) but also to continuous ones, such as those commonly used in summarization and MT. For different generation tasks, we employ task-specific response quality measures: accuracy for MMLU and GSM8k, COMET (Rei et al., 2020) for machine translation, and AlignScore (Zha et al., 2023) for the remaining tasks. For summarization tasks, we use AlignScore between the output summary and the input document to measure the factuality of the generation.

# B. Results of Ablation Studies

## B.1. Aggregation Strategies and Hyperparameter Sensitivity

Tables 5 to 10 present the performance of the RAUQ method using various aggregation functions for token-level confidence scores, layer-wise uncertainty scores, recurrent formulas for computing token-level confidence scores, normalization variants of the attention term, and token-level confidence signals, respectively. For the comparison of confidence signals, we use representative $\alpha$ values from the hyperparameter analysis in Section 5.3: $\alpha = 0.2$ for probability-based RAUQ and $\alpha = 0.5$ for entropy-based RAUQ on the base model, and $\alpha = 0.6$ and $\alpha = 0.9$, respectively, on the instruction-tuned model.

*Table 5.* PRR↑ for Llama-3.1 8B model for various aggregation function of token-level confidence scores. The best method is in **bold**, the second best is underlined.

| Token Aggregation | XSum | SamSum | CNN | WMT14 | WMT19 | MedQUAD | TruthfulQA | CoQA | SciQ | TriviaQA | MMLU | GSM8k | Mean |
|---|---|---|---|---|---|---|---|---|---|---|---|---|---|
| $-\frac{1}{N}\sum_{i=1}^{N}\mathbf{c}_l^t(y_i)$ | **.375** | .419 | **.460** | .359 | .485 | .140 | .304 | .259 | **.511** | **.534** | .526 | **.339** | .393 |
| $-\mathrm{median}_{i=1}^{N}\mathbf{c}_l^t(y_i)$ | .267 | .403 | .437 | .249 | .340 | .154 | .317 | .234 | .430 | .432 | .635 | .253 | .346 |
| $-\sum_{i=1}^{N}\log\mathbf{c}_l^t(y_i)$ | .027 | -.045 | .325 | .224 | .242 | .107 | .035 | .114 | .202 | .300 | **.658** | .213 | .198 |
| $-\frac{1}{N}\sum_{i=1}^{N}\log\mathbf{c}_l^t(y_i)$ | .370 | **.464** | .452 | **.394** | **.509** | **.249** | **.364** | **.265** | .506 | .522 | .549 | .323 | **.413** |

*Table 6.* PRR↑ for Llama-3.1 8B model for various aggregation function of layer-wise uncertainty scores. The best method is in **bold**, the second best is underlined.

| Layer Aggregation | XSum | SamSum | CNN | WMT14 | WMT19 | MedQUAD | TruthfulQA | CoQA | SciQ | TriviaQA | MMLU | GSM8k | Mean |
|---|---|---|---|---|---|---|---|---|---|---|---|---|---|
| $\frac{1}{|\mathcal{L}|}\sum_{l\in\mathcal{L}}\mathbf{u}_l(y)$ | **.384** | .419 | **.475** | .389 | .519 | .154 | .345 | **.274** | .496 | **.535** | .529 | .337 | .404 |
| $\mathrm{median}_{l\in\mathcal{L}}\,\mathbf{u}_l(y)$ | .378 | .426 | .471 | .388 | **.526** | .246 | .351 | .267 | .502 | .532 | .532 | **.340** | .412 |
| $\max_{l\in\mathcal{L}}\,\mathbf{u}_l(y)$ | .370 | **.464** | .452 | **.394** | .509 | **.249** | **.364** | .265 | **.506** | .522 | **.549** | .323 | **.413** |

*Table 7.* PRR↑ for Llama-3.1 8B model for various function for recurrent calculation of confidence scores $\mathbf{c}_l(y_i)$ in Equation (2). The best method is in **bold**, the second best is underlined.

| Recurrent Formula | XSum | SamSum | CNN | WMT14 | WMT19 | MedQUAD | TruthfulQA | CoQA | SciQ | TriviaQA | MMLU | GSM8k | Mean |
|---|---|---|---|---|---|---|---|---|---|---|---|---|---|
| $\alpha\cdot P(y_i\mid\mathbf{x},y_{<i})+(1-\alpha)\cdot\mathbf{c}_l(y_{i-1})$ | .330 | .383 | .393 | .238 | .313 | **.273** | .224 | .267 | .514 | .517 | .475 | .279 | .330 |
| $\alpha\cdot P(y_i\mid\mathbf{x},y_{<i})+(1-\alpha)\cdot a_{i,i-1}^{l\,\mathbf{h}_i}$ | **.412** | .387 | .457 | .332 | .436 | .205 | .322 | .257 | .485 | .517 | .550 | .305 | .388 |
| $\alpha\cdot P(y_i\mid\mathbf{x},y_{<i})+(1-\alpha)\cdot a_{i,i-1}^{l\,\mathbf{h}_i}\cdot P(y_{i-1}\mid\mathbf{x},y_{<i-1})$ | .399 | .421 | **.461** | .370 | .472 | .235 | .336 | **.279** | .456 | .517 | .532 | .318 | .399 |
| $P(y_i\mid\mathbf{x},y_{<i})\cdot a_{i,i-1}^{l\,\mathbf{h}_i}$ | .394 | .327 | .459 | .226 | .337 | .149 | .251 | .161 | .330 | .330 | **.645** | .255 | .322 |
| $\alpha\cdot P(y_i\mid\mathbf{x},y_{<i})+(1-\alpha)\cdot a_{i,i-1}^{l\,\mathbf{h}_i}\cdot\mathbf{c}_l(y_{i-1})$ | .370 | **.464** | .452 | **.394** | **.509** | .249 | **.364** | .265 | **.506** | **.522** | .549 | **.323** | **.413** |

*Table 8.* PRR↑ for Llama-3.1 8B with various normalizations of attention in RAUQ, using all previous tokens or the last $K$ tokens. Here, $\bar{a}_i^l = \frac{1}{i-1}\sum_{j=2}^{i}a_{j,j-1}^{l\,\mathbf{h}_l}$, and $\bar{a}_{i,K}^l = \frac{1}{K}\sum_{j=i-K+1}^{i}a_{j,j-1}^{l\,\mathbf{h}_l}$, with $K=5$. The best method is in **bold**, the second best is underlined.

| Recurrent Formula | XSum | SamSum | CNN | WMT14 | WMT19 | MedQUAD | TruthfulQA | CoQA | SciQ | TriviaQA | MMLU | GSM8k | Mean |
|---|---|---|---|---|---|---|---|---|---|---|---|---|---|
| $\alpha\cdot P(y_i\mid\mathbf{x},y_{<i})+(1-\alpha)\cdot(a_{i,i-1}^{l\,\mathbf{h}_i}-\bar{a}_{i,K}^l)\cdot\mathbf{c}_l(y_{i-1})$ | **.376** | .339 | .096 | .353 | .439 | **.292** | **.378** | .257 | .473 | .533 | **.550** | .304 | .366 |
| $\alpha\cdot P(y_i\mid\mathbf{x},y_{<i})+(1-\alpha)\cdot(a_{i,i-1}^{l\,\mathbf{h}_i}-\bar{a}_i^l)\cdot\mathbf{c}_l(y_{i-1})$ | .364 | .332 | .103 | .345 | .440 | .266 | .369 | .255 | .483 | .532 | **.550** | .296 | .361 |
| $\alpha\cdot P(y_i\mid\mathbf{x},y_{<i})+(1-\alpha)\cdot\frac{a_{i,i-1}^{l\,\mathbf{h}_i}}{\bar{a}_i^l+\varepsilon}\cdot\mathbf{c}_l(y_{i-1})$ | .349 | .253 | .072 | .164 | .283 | .281 | .273 | .243 | .385 | **.535** | .460 | .173 | .289 |
| $\alpha\cdot P(y_i\mid\mathbf{x},y_{<i})+(1-\alpha)\cdot a_{i,i-1}^{l\,\mathbf{h}_i}\cdot\mathbf{c}_l(y_{i-1})$ | .370 | **.464** | **.452** | **.394** | **.509** | .249 | .364 | **.265** | **.506** | .522 | .549 | **.323** | **.413** |

*Table 9.* PRR↑ for the Llama-3.1 8B model using probability-based and entropy-based token-level confidence signals in RAUQ. The best method is in **bold**.

| UQ Method | XSum | SamSum | CNN | WMT14 | WMT19 | MedQUAD | TruthfulQA | CoQA | SciQ | TriviaQA | MMLU | GSM8k | Mean |
|---|---|---|---|---|---|---|---|---|---|---|---|---|---|
| RAUQ | .433 | .380 | **.157** | .418 | .481 | **.265** | .379 | **.248** | .509 | **.543** | **.536** | .310 | .388 |
| RAUQ (Entropy) | **.436** | **.395** | .146 | **.487** | **.541** | .244 | **.385** | .230 | **.525** | .531 | .411 | **.365** | **.391** |

*Table 10.* PRR↑ for the Llama-3.1 8B-Instruct model using probability-based and entropy-based token-level confidence signals in RAUQ. The best method is in **bold**.

| UQ Method | XSum | SamSum | CNN | WMT14 | WMT19 | MedQUAD | CoQA | SciQ | TriviaQA | MMLU | GSM8k | Mean |
|---|---|---|---|---|---|---|---|---|---|---|---|---|
| RAUQ | **.119** | .208 | .114 | .384 | .504 | .317 | .344 | .545 | .547 | **.543** | .408 | .367 |
| RAUQ (Entropy) | .116 | **.215** | **.118** | **.455** | **.555** | **.381** | **.359** | **.598** | **.610** | .540 | **.453** | **.400** |

## B.2. Layers and Heads Selection

**Layer selection analysis.** Table 11 presents the performance of the RAUQ method across various layer subsets. We compare RAUQ using individual layers, all layers, and aggregated middle layers. In our experiments, we consistently use the same range of layers – from the first third to the second third of the model (e.g., layers 10 to 22 for Llama-3.1 8B) without any task- or model-specific tuning.

The results indicate that although certain layers (e.g., the 25th or 30th) perform better on specific tasks, they tend to underperform on average. While the selection of the optimal layer (e.g., 22nd for Llama-3.1 8B) can slightly improve overall performance, it requires supervision, whereas our method is designed to be fully unsupervised. Using all layers instead of only the middle layers yields only a marginal decrease in performance for RAUQ on Llama-3.1 8B, with an average drop of just 0.003 in PRR. Therefore, while this modification can slightly enhance results, it is not a critical component of our method.

*Table 11.* PRR↑ for Llama-3.1 8B model for various layer subsets $\mathcal{L}$ in Equation (4). The best method is in **bold**, the second best is underlined.

| Layer Subset | WMT14 | WMT19 | MedQUAD | TruthfulQA | CoQA | SciQ | Mean |
|---|---|---|---|---|---|---|---|
| RAUQ (22nd layer) | **.412** | **.529** | .237 | .354 | .267 | **.514** | **.385** |
| RAUQ (25th layer) | .359 | .519 | .244 | **.382** | .262 | .462 | .371 |
| RAUQ (30th layer) | .272 | .433 | .168 | .326 | **.268** | .456 | .320 |
| RAUQ (5 middle layers, 14–18) | .388 | .502 | .240 | .365 | .258 | .510 | .377 |
| RAUQ (All layers) | .386 | .516 | **.244** | .366 | .260 | .490 | .377 |
| RAUQ | .394 | .509 | .241 | .364 | .265 | .506 | .380 |

**Head selection analysis.** Tables 12 and 13 present an analysis of the selected attention heads for the WMT14 De-En and CoQA datasets using Llama-3.1 8B. We report the top-3 heads based on their selection frequency according to our criterion, along with the corresponding percentages.

First, the tables show that in most cases, the most frequently selected head accounts for more than 90% of instances, indicating high stability in head selection. Even in layers where head selection is less consistent, the top-3 heads still cover more than 90% of cases, suggesting that the model typically chooses similar heads across inputs within the same task.

Second, when comparing selected heads across the two datasets, we observed substantial overlap. For example, in layers 10, 12, 13, 15, 16 and 20, the selected heads are fully aligned, reflecting strong cross-task consistency. Overall, while some variation exists, the same heads generally provide the most informative signals used in the RAUQ method, highlighting both intra-task and cross-task stability.

*Table 12.* The top three most frequently selected attention heads per layer in the Llama-3.1 8B model on the WMT14 dataset with its selection frequency according to our criterion.

| Attention Head | Layer 10 | Layer 11 | Layer 12 | Layer 13 | Layer 14 | Layer 15 | Layer 16 | Layer 17 | Layer 18 | Layer 19 | Layer 20 | Layer 21 | Layer 22 |
|---|---|---|---|---|---|---|---|---|---|---|---|---|---|
| Top-1 head | 10 (87.5%) | 10 (99.2%) | 12 (100.0%) | 28 (100.0%) | 19 (84.2%) | 6 (99.9%) | 30 (99.7%) | 12 (83.0%) | 29 (46.2%) | 11 (97.2%) | 3 (99.7%) | 10 (50.9%) | 9 (99.3%) |
| Top-2 head | 0 (12.3%) | 16 (0.6%) | - | - | 14 (12.9%) | 24 (0.1%) | 22 (0.4%) | 22 (16.4%) | 14 (29.0%) | 8 (2.2%) | 0 (0.3%) | 9 (26.7%) | 19 (0.3%) |
| Top-3 head | 18 (0.1%) | 12 (0.1%) | - | - | 8 (2.5%) | - | - | 6 (0.3%) | 26 (11.1%) | 10 (0.5%) | - | 3 (15.4%) | 11 (0.2%) |

*Table 13.* The top three most frequently selected attention heads per layer in the Llama-3.1 8B model on the CoQA dataset with its selection frequency according to our criterion.

| Attention Head | Layer 10 | Layer 11 | Layer 12 | Layer 13 | Layer 14 | Layer 15 | Layer 16 | Layer 17 | Layer 18 | Layer 19 | Layer 20 | Layer 21 | Layer 22 |
|---|---|---|---|---|---|---|---|---|---|---|---|---|---|
| Top-1 head | 0 (95.2%) | 10 (76.8%) | 12 (100.0%) | 28 (100.0%) | 16 (27.0%) | 6 (95.0%) | 30 (91.3%) | 12 (76.7%) | 29 (54.6%) | 11 (61.5%) | 3 (66.0%) | 10 (74.7%) | 9 (64.2%) |
| Top-2 head | 10 (4.3%) | 23 (7.3%) | - | - | 8 (26.5%) | 24 (2.8%) | 22 (8.6%) | 6 (18.9%) | 25 (17.5%) | 8 (17.5%) | 0 (23.8%) | 8 (7.8%) | 19 (26.5%) |
| Top-3 head | 18 (0.4%) | 31 (5.1%) | - | - | 14 (17.1%) | 4 (0.7%) | 9 (0.1%) | 30 (1.2%) | 26 (15.5%) | 23 (3.3%) | 27 (3.8%) | 9 (5.9%) | 18 (2.4%) |

**Experiments with a single optimal head.** Table 14 presents an analysis in which a single optimal head per layer is selected for all inputs determined on a small held-out validation set of 100 instances per task. The results show that the gains from such precise per-dataset head selection are marginal and the average performance across all datasets remains effectively similar. This indicates that retaining dynamic unsupervised head selection as part of the algorithm fully removes the need for any precise task-specific adjustments and already achieves near-optimal performance. This design choice also ensures that the method remains entirely unsupervised, requires no validation data, and is seamlessly plug-and-play for any new LLM or task.

*Table 14.* PRR↑ for Llama-3.1 8B model for RAUQ with dynamic head selection per input and with a single optimal head per layer, fixed across all inputs. The best method is in **bold**.

| UQ Method | XSum | SamSum | CNN | WMT14 | WMT19 | MedQUAD | TruthfulQA | CoQA | SciQ | TriviaQA | MMLU | GSM8k | Mean |
|---|---|---|---|---|---|---|---|---|---|---|---|---|---|
| RAUQ | **.384** | .423 | .189 | .406 | **.488** | **.317** | **.399** | .248 | **.506** | **.548** | .513 | .323 | **.395** |
| RAUQ (Single Head) | .382 | **.426** | **.195** | **.407** | .481 | .303 | .386 | **.257** | .494 | .544 | **.528** | **.325** | .394 |

# C. Additional Experimental Results

## C.1. Experiments with Diverse LLM Sizes

To demonstrate that RAUQ generalizes effectively to both larger and smaller LLMs, we have conducted additional experiments using SmolLM-2 360M, Llama-3.2 1B and Llama-3.1 70B. The results are presented in Table 15. For models with parameters $\leq$ 1B, we exclude MMLU, GSM8k, and MedQUAD due to their near-zero performance on these tasks.

The results show that RAUQ achieves the best performance on QA and MT for models with $\leq$ 1B parameters and on MT for the 70B model. For QA on the 70B model, RAUQ is second overall. Across all tasks and model sizes, RAUQ surpasses the second-best method by an average of 2% of PRR. These results highlight the strong generalization capability of RAUQ in a wide range of model sizes.

*Table 15.* Mean PRR↑ across tasks for the evaluated LLMs (≤1B and 70B). Warmer color indicates better results.

| UQ Method | SmolLM-2 360M | | | Llama-3.2 1B | | | Llama-3.1 70B | | | Mean |
| --- | --- | --- | --- | --- | --- | --- | --- | --- | --- | --- |
| | QA | Summ | MT | QA | Summ | MT | QA | Summ | MT | |
| MSP | .360 | .449 | .330 | .324 | .507 | .351 | .364 | .128 | .447 | .362 |
| Perplexity | .371 | .330 | .487 | .310 | .392 | .427 | .323 | .245 | .335 | .358 |
| CCP | .281 | **.457** | .361 | .283 | **.517** | .328 | .350 | .135 | .387 | .344 |
| Attention Score | .071 | .004 | .120 | .051 | .033 | .103 | .053 | .045 | .213 | .077 |
| Simple Focus | .401 | .429 | .410 | **.370** | .488 | .424 | .380 | .128 | .435 | .385 |
| DegMat NLI Score entail. | .342 | .059 | .227 | .305 | .078 | .287 | .380 | .091 | .273 | .227 |
| Ecc. NLI Score entail. | .209 | -.013 | .169 | .225 | -.012 | .293 | .330 | -.003 | .298 | .166 |
| EVL NLI Score entail. | .333 | .055 | .216 | .298 | .072 | .268 | .369 | .091 | .265 | .219 |
| Lexical Similarity Rouge-L | .290 | -.013 | .193 | .255 | .074 | .337 | .362 | .089 | .332 | .213 |
| EigenScore | .173 | .068 | .061 | .145 | .029 | .301 | .296 | .044 | .325 | .160 |
| LUQ | .337 | .076 | .242 | .279 | .118 | .263 | .376 | .139 | .254 | .232 |
| Semantic Entropy | .201 | .067 | .227 | .187 | .084 | .268 | .309 | .069 | .373 | .198 |
| SAR | .343 | .095 | .348 | .295 | .091 | .408 | .382 | .106 | .372 | .271 |
| Semantic Density | .357 | .209 | .259 | .348 | .217 | .285 | **.385** | .100 | .239 | .267 |
| RAUQ | **.425** | .356 | **.490** | .356 | .423 | **.495** | .360 | **.245** | **.457** | **.401** |

## C.2. Experiments with Instruction-Tuned Models

To further evaluate the applicability of RAUQ beyond base LLMs, we conducted additional experiments with instruction-tuned models: Llama-3.1 8B-Instruct (Dubey et al., 2024) and GPT-OSS 20B (OpenAI, 2025). The results are presented in Table 16. In these experiments, we used the entropy-based variant of RAUQ, while keeping the same attention-head selection and recurrent aggregation mechanism.

The results show that RAUQ achieves the best overall performance for instruction-tuned models and outperforms the considered baselines in most cases. Additionally, our experiments with GPT-OSS 20B suggest that similar hallucination-associated attention patterns also hold for Mixture-of-Experts (MoE) models (Shazeer et al., 2017).

## C.3. Experiments Using the ROC-AUC Metric

The results evaluated using the ROC-AUC metric are presented in Table 17. For all generation quality metrics, except accuracy, we compute scores by thresholding the original continuous values to obtain discrete versions of the quality metrics. The thresholds were empirically determined as follows: 0.5 for QA and Summ, and 0.85 for MT.

We observe similar trends to those with the PRR metric. RAUQ substantially outperforms all other methods for summarization and MT tasks. For QA, RAUQ is the best method for Llama-3.1 8B and Falcon-3 10B, while performing comparably to computationally intensive sampling-based approaches for other models. Overall, RAUQ achieves a 0.6% improvement over the second-best method (Perplexity) across all evaluated models.

*Table 16.* PRR↑ for instruction-tuned LLMs. Warmer color indicates better results.

| UQ Method | GPT-OSS 20B | | | Llama-3.1 8B-Instruct | | | Mean |
|---|---|---|---|---|---|---|---|
| | QA | Summ | MT | QA | Summ | MT | |
| MSP | .387 | .220 | .422 | .408 | .184 | .383 | .334 |
| Perplexity | .388 | .162 | .471 | .419 | .139 | .422 | .334 |
| MTE | .399 | .180 | **.532** | .458 | .144 | .478 | .365 |
| CCP | .355 | .179 | .347 | .395 | .175 | .341 | .299 |
| Attention Score | .111 | **.321** | .149 | .089 | **.230** | .137 | .173 |
| Simple Focus | .380 | .142 | .427 | .422 | .119 | .374 | .311 |
| DegMat NLI Score entail. | .395 | .090 | .372 | .425 | .101 | .356 | .290 |
| Ecc. NLI Score entail. | .347 | .067 | .384 | .378 | .060 | .377 | .269 |
| EVL NLI Score entail. | .383 | .086 | .374 | .415 | .093 | .357 | .285 |
| Lexical Similarity Rouge-L | .381 | .120 | .418 | .446 | .128 | .399 | .316 |
| EigenScore | .305 | .068 | .391 | .348 | .107 | .437 | .276 |
| LUQ | .389 | .107 | .244 | .432 | .121 | .306 | .267 |
| Semantic Entropy | .373 | .194 | .437 | .395 | .152 | .416 | .328 |
| SAR | .403 | .162 | .481 | .453 | .138 | .467 | .350 |
| Semantic Density | .194 | .168 | .359 | .241 | .138 | .360 | .243 |
| RAUQ | **.406** | .196 | .525 | **.490** | .150 | **.505** | **.379** |

*Table 17.* Mean ROC-AUC↑ across tasks for the evaluated LLMs. Warmer color indicates better results.

| UQ Method | Llama-3.1 8B | | | Qwen-2.5 7B | | | Gemma-2 9B | | | Falcon-3 10B | | | Mean |
|---|---|---|---|---|---|---|---|---|---|---|---|---|---|
| | QA | Summ | MT | QA | Summ | MT | QA | Summ | MT | QA | Summ | MT | |
| MSP | .711 | .718 | .686 | .700 | .559 | .685 | .746 | .735 | .683 | .721 | .583 | .688 | .685 |
| Perplexity | .701 | .812 | .690 | .705 | **.661** | .713 | .735 | .766 | .699 | .713 | **.606** | .715 | .710 |
| CCP | .685 | .705 | .648 | .668 | .575 | .658 | .729 | .731 | .646 | .703 | .569 | .657 | .665 |
| Attention Score | .497 | .552 | .553 | .522 | .530 | .540 | .519 | .536 | .543 | .534 | .590 | .539 | .538 |
| Focus | .698 | .746 | .663 | .642 | .612 | .682 | .747 | .738 | .684 | .699 | .577 | .672 | .680 |
| Simple Focus | .718 | .730 | .694 | .703 | .588 | .700 | **.753** | .723 | .706 | .724 | .543 | .691 | .689 |
| DegMat NLI Score entail. | .676 | .591 | .618 | .691 | .604 | .637 | .692 | .612 | .636 | .700 | .581 | .620 | .638 |
| Ecc. NLI Score entail. | .659 | .498 | .630 | .682 | .510 | .650 | .678 | .535 | .642 | .688 | .546 | .648 | .614 |
| EVL NLI Score entail. | .668 | .590 | .610 | .688 | .602 | .630 | .690 | .607 | .632 | .703 | .583 | .612 | .635 |
| Lexical Similarity Rouge-L | .659 | .605 | .660 | .687 | .594 | .677 | .684 | .620 | .668 | .673 | .559 | .646 | .644 |
| EigenScore | .643 | .533 | .629 | .675 | .549 | .655 | .658 | .592 | .614 | .662 | .527 | .623 | .613 |
| LUQ | .667 | .633 | .618 | .688 | .627 | .613 | .690 | .644 | .629 | .687 | .570 | .599 | .639 |
| Semantic Entropy | .661 | .583 | .658 | .680 | .544 | .665 | .683 | .595 | .661 | .706 | .579 | .666 | .640 |
| SAR | .696 | .627 | .692 | **.708** | .590 | .710 | .723 | .670 | .710 | .712 | .569 | .670 | .673 |
| Semantic Density | .694 | .582 | .628 | .705 | .572 | .635 | .711 | .611 | .617 | .721 | .583 | .624 | .640 |
| RAUQ | **.724** | **.815** | **.713** | .705 | .629 | **.715** | .752 | **.772** | **.718** | **.726** | .597 | **.727** | **.716** |

## C.4. Comparison with Supervised Methods

We compare our method against several state-of-the-art supervised methods that leverage hidden states or attention weights: Factoscope (He et al., 2024a), SAPLMA (Azaria & Mitchell, 2023), MIND (Su et al., 2024), Sheeps (CH-Wang et al., 2024), LookBack Lens (Chuang et al., 2024), SATRMD+MSP (Vazhentsev et al., 2025b), and TAD (Vazhentsev et al., 2025a). We evaluate these methods in two scenarios: in-domain, where the model is trained directly on the target task, and out-of-domain, where the model is trained on all datasets except one, which is held out for testing. Tables 18 and 19 show the performance of supervised methods in the in-domain and out-of-domain settings, respectively.

The results show that in the in-domain experimental setup, supervised methods leveraging attention-based features, such as TAD and LookBackLens, outperform the RAUQ method. Methods that leverage hidden states, such as MIND and Sheeps, achieve performance comparable to RAUQ on average, but underperform on the CNN and WMT19 datasets. In contrast, in the out-of-domain experimental setup, RAUQ substantially outperforms on average all supervised methods, which experience a substantial performance drop. Our method, however, maintains consistent performance due to its unsupervised nature.

Overall, RAUQ approaches the performance of most supervised methods in in-domain settings, underperforming only those

based on attention, while requiring no access to the training dataset. In out-of-domain settings, RAUQ demonstrates a strong advantage, substantially outperforming all supervised approaches.

*Table 18.* Comparison with supervised methods by PRR↑ for the Llama-3.1 8B model in the in-domain setup across each dataset. The best method is in **bold**, the second best is underlined. Warmer color indicates better results.

| UQ Method | XSum | SamSum | CNN | WMT19 | TruthfulQA | CoQA | SciQ | TriviaQA | MMLU | GSM8k | Mean |
|---|---|---|---|---|---|---|---|---|---|---|---|
| Factoscope | .292 | .064 | -.020 | .120 | .065 | .033 | .313 | .363 | .585 | .121 | .194 |
| SAPLMA | .288 | .382 | .056 | .548 | .277 | -.002 | .399 | .399 | .456 | .358 | .316 |
| MIND | .437 | .361 | .178 | .451 | .411 | .263 | .499 | .517 | **.727** | .570 | .441 |
| Sheeps | .510 | .466 | .380 | .509 | .349 | **.423** | .552 | .594 | .723 | **.604** | .511 |
| LookBackLens | .528 | .441 | .279 | **.613** | .462 | .341 | .542 | .497 | .718 | .525 | .495 |
| SATRMD+MSP | .494 | .495 | .248 | .475 | .448 | .333 | **.581** | .561 | .704 | .528 | .487 |
| TAD | **.550** | **.535** | .444 | .592 | **.463** | .392 | .488 | **.632** | .724 | .557 | **.538** |
| RAUQ | .370 | .464 | **.452** | .509 | .364 | .265 | .506 | .522 | .549 | .323 | .432 |

*Table 19.* Comparison with supervised methods by PRR↑ for the Llama-3.1 8B model in the out-of-domain setup across each dataset. The best method is in **bold**, the second best is underlined. Warmer color indicates better results.

| UQ Method | XSum | SamSum | CNN | WMT19 | TruthfulQA | CoQA | SciQ | TriviaQA | MMLU | GSM8k | Mean |
|---|---|---|---|---|---|---|---|---|---|---|---|
| Factoscope | .105 | .050 | -.065 | .083 | .036 | .014 | .084 | -.017 | .007 | -.040 | .026 |
| SAPLMA | -.035 | .049 | -.009 | -.029 | -.056 | -.020 | -.010 | .224 | -.000 | .152 | .027 |
| MIND | -.077 | .185 | .074 | .158 | .281 | .112 | .166 | .222 | .352 | .316 | .179 |
| Sheeps | .122 | .101 | -.056 | .013 | **.410** | .184 | .365 | .223 | .535 | .310 | .221 |
| LookBackLens | .171 | .197 | .000 | -.018 | .220 | .116 | .285 | .178 | .316 | .189 | .166 |
| SATRMD+MSP | .362 | .098 | **.477** | .364 | .108 | .142 | .190 | .170 | **.572** | .307 | .279 |
| TAD | .269 | .176 | -.101 | .087 | .224 | .143 | .251 | .394 | .432 | **.323** | .220 |
| RAUQ | **.370** | **.464** | .452 | **.509** | .364 | **.265** | **.506** | **.522** | .549 | .323 | **.432** |

## C.5. Experiments with Interpretability Scores

To assess the flexibility and generalization of RAUQ beyond standard LLM architectures with attention layers, we evaluate its performance when original attention weights are replaced with alternative interpretability scores, such as Layer Integrated Gradients (LIG) (Sundararajan et al., 2017).

We conducted an experiment using the Llama-3.1 8B model, where we replaced the original attention weights with scores derived from Layer Integrated Gradients computed on the output projection layer following the attention module. We manually partition this linear layer in each transformer block into equal segments corresponding to a synthetic division across attention heads, and compute interpretability scores for each segment using Layer Integrated Gradients. This procedure yields matrices analogous to attention weights, preserving the same number of "heads" and layers. We then apply these matrices within the RAUQ method without any modification.

The results indicate that RAUQ (LIG) performs comparably to the original RAUQ, with only a negligible performance degradation of 0.4% PRR on average across datasets. These experiments further illustrate that original attention can be effectively substituted with alternative interpretability scores, enabling the application of RAUQ to models without attention mechanisms or with non-standard attention architectures.

*Table 20.* PRR↑ for Llama-3.1 8B model for RAUQ with original attention weights and with Layer Integrated Gradients (LIG) instead of attention weights. The best method is in **bold**.

| UQ Method | WMT14 | WMT19 | TruthfulQA | CoQA | SciQ | TriviaQA | MMLU | Mean |
|---|---|---|---|---|---|---|---|---|
| RAUQ | **.394** | .509 | **.364** | **.265** | **.506** | **.522** | **.549** | **.444** |
| RAUQ (LIG) | .389 | **.512** | .362 | .264 | .489 | .515 | .547 | .440 |

# D. Detailed Experimental Results

The detailed experimental results across each considered dataset are presented in Tables 21 to 24 for Llama-3.1 8B, Qwen-2.5 7B, Gemma-2 9B and Falcon-3 10B models, respectively.

*Table 21.* Detailed PRR↑ for the Llama-3.1 8B model across each dataset. The best method is in **bold**, the second best is underlined. Warmer color indicates better results.

| UQ Method | XSum | SamSum | CNN | WMT14 | WMT19 | MedQUAD | TruthfulQA | CoQA | SciQ | TriviaQA | MMLU | GSM8k | Mean |
|---|---|---|---|---|---|---|---|---|---|---|---|---|---|
| MSP | .313 | .050 | **.525** | .335 | .459 | .091 | .242 | .262 | .459 | .527 | _.535_ | .310 | .342 |
| Perplexity | **.370** | _.456_ | .431 | .344 | .416 | **.249** | _.377_ | .259 | .244 | .506 | .492 | .303 | _.370_ |
| CCP | .347 | .059 | _.514_ | .317 | .363 | .038 | .080 | .210 | .351 | .562 | .446 | .306 | .299 |
| Attention Score | .036 | .083 | .258 | .176 | .179 | -.295 | .081 | -.028 | -.142 | .067 | .209 | .209 | .069 |
| Focus | .326 | .281 | .399 | .306 | .416 | .137 | **.380** | .211 | .422 | .507 | .305 | .278 | .331 |
| Simple Focus | .272 | .193 | .454 | _.358_ | _.472_ | .074 | .187 | .281 | _.486_ | .545 | .516 | .302 | .345 |
| DegMat NLI Score entail. | .033 | .147 | .173 | .193 | .285 | .146 | .226 | _.316_ | .429 | **.583** | .239 | .203 | .248 |
| Ecc. NLI Score entail. | .011 | -.004 | -.031 | .229 | .340 | .102 | .145 | .293 | .380 | .530 | .231 | .235 | .205 |
| EVL NLI Score entail. | .035 | .144 | .164 | .183 | .252 | .137 | .234 | .314 | .371 | _.577_ | .230 | .188 | .236 |
| Lexical Similarity Rouge-L | .081 | .122 | .190 | .246 | .403 | -.017 | .110 | .277 | .378 | .491 | .242 | .273 | .233 |
| EigenScore | .036 | .130 | .069 | .252 | .318 | -.010 | .079 | .263 | .355 | .462 | .192 | .283 | .202 |
| LUQ | .141 | .221 | .156 | .204 | .224 | .101 | .235 | .303 | .394 | .570 | .249 | .158 | .246 |
| Semantic Entropy | .025 | .105 | .222 | .252 | .379 | .093 | .107 | .232 | .347 | .479 | .157 | **.366** | .230 |
| SAR | .060 | .224 | .227 | .306 | .435 | .107 | .181 | .297 | .439 | .552 | .275 | .320 | .285 |
| Semantic Density | .158 | .154 | .148 | .233 | .295 | .175 | .302 | **.380** | .448 | .571 | .237 | .197 | .275 |
| RAUQ | _.370_ | **.464** | .452 | **.394** | **.509** | _.241_ | .364 | .265 | **.506** | .522 | .549 | _.323_ | **.413** |

*Table 22.* Detailed PRR↑ for the Qwen-2.5 7B model across each dataset. The best method is in **bold**, the second best is underlined. Warmer color indicates better results.

| UQ Method | XSum | SamSum | CNN | WMT14 | WMT19 | MedQUAD | TruthfulQA | CoQA | SciQ | TriviaQA | MMLU | GSM8k | Mean |
|---|---|---|---|---|---|---|---|---|---|---|---|---|---|
| MSP | .088 | -.003 | **.368** | .286 | .451 | .030 | -.101 | .291 | **.551** | _.610_ | **.654** | .268 | .291 |
| Perplexity | _.242_ | **.289** | .229 | _.346_ | .466 | **.131** | .156 | .270 | .385 | .601 | .400 | .456 | _.331_ |
| CCP | **.243** | .021 | _.294_ | .266 | .388 | .015 | -.089 | .215 | .468 | .596 | .412 | .281 | .259 |
| Attention Score | .037 | .103 | .250 | .136 | .149 | .022 | -.023 | .007 | -.105 | .078 | .157 | .131 | .078 |
| Focus | .214 | .149 | .196 | .308 | .452 | .123 | .137 | .249 | .462 | .568 | .037 | .273 | .264 |
| Simple Focus | .117 | .086 | .205 | .302 | _.496_ | .021 | .037 | .321 | .536 | **.620** | .550 | .310 | .300 |
| DegMat NLI Score entail. | .141 | .178 | .145 | .217 | .332 | .122 | .293 | .329 | _.540_ | .574 | .235 | .402 | .292 |
| Ecc. NLI Score entail. | -.058 | .044 | .021 | .243 | .368 | .107 | .151 | .294 | .535 | .543 | .237 | .386 | .239 |
| EVL NLI Score entail. | .141 | .183 | .138 | .196 | .294 | .122 | _.294_ | .329 | .519 | .571 | .236 | .372 | .283 |
| Lexical Similarity Rouge-L | .119 | .161 | .112 | .284 | .370 | .075 | .141 | .297 | .507 | .531 | .274 | _.511_ | .282 |
| EigenScore | .079 | .034 | .071 | .231 | .374 | .018 | -.003 | .281 | .510 | .500 | .243 | **.537** | .240 |
| LUQ | .224 | _.260_ | .104 | .161 | .265 | .096 | **.340** | _.337_ | .449 | .580 | .321 | .331 | .289 |
| Semantic Entropy | .021 | .109 | .146 | .268 | .366 | .073 | .058 | .265 | .491 | .536 | .165 | .380 | .240 |
| SAR | .128 | .186 | .145 | .340 | .445 | .088 | .196 | .318 | .526 | .585 | .288 | .459 | .309 |
| Semantic Density | .084 | .156 | .092 | .225 | .358 | .095 | .285 | **.386** | .514 | .603 | .203 | .381 | .282 |
| RAUQ | .180 | .206 | .254 | _.344_ | **.533** | _.123_ | -.020 | .252 | .499 | .608 | _.584_ | .458 | **.335** |

*Table 23.* Detailed PRR↑ for the Gemma-2 9B model across each dataset. The best method is in **bold**, the second best is underlined. Warmer color indicates better results.

| UQ Method | XSum | SamSum | CNN | WMT14 | WMT19 | MedQUAD | TruthfulQA | CoQA | SciQ | TriviaQA | MMLU | GSM8k | Mean |
|---|---|---|---|---|---|---|---|---|---|---|---|---|---|
| MSP | _.333_ | .095 | **.574** | .279 | .484 | .004 | .152 | .310 | _.501_ | .649 | _.599_ | .310 | .357 |
| Perplexity | .329 | _.308_ | .488 | .362 | .449 | _.397_ | .240 | .314 | .234 | **.660** | .578 | .256 | .385 |
| CCP | **.407** | .061 | _.566_ | .270 | .369 | .028 | .092 | .277 | .385 | .633 | .550 | .339 | .332 |
| Attention Score | -.043 | .061 | .291 | .131 | .161 | -.150 | .083 | -.016 | -.112 | .075 | .300 | .268 | .087 |
| Focus | .276 | .308 | .436 | .305 | .465 | **.514** | .230 | .289 | .434 | .619 | .563 | .265 | _.392_ |
| Simple Focus | .258 | .169 | .538 | .324 | _.521_ | .170 | .238 | _.335_ | **.523** | _.656_ | .570 | .280 | .382 |
| DegMat NLI Score entail. | .061 | .232 | .120 | .206 | .312 | .167 | .141 | .312 | .422 | .619 | .401 | .293 | .274 |
| Ecc. NLI Score entail. | -.000 | .072 | -.012 | .237 | .343 | .037 | .132 | .299 | .419 | .569 | .399 | .228 | .227 |
| EVL NLI Score entail. | .062 | .231 | .105 | .202 | .302 | .176 | .159 | .304 | .389 | .615 | .398 | .284 | .269 |
| Lexical Similarity Rouge-L | .059 | .168 | .257 | .279 | .404 | -.035 | .113 | .319 | .395 | .585 | .418 | _.346_ | .276 |
| EigenScore | .016 | .082 | .221 | .204 | .249 | -.024 | .132 | .270 | .359 | .519 | .371 | .241 | .220 |
| LUQ | .199 | .247 | .172 | .242 | .276 | .222 | .250 | .301 | .342 | .618 | .440 | .237 | .295 |
| Semantic Entropy | .013 | .101 | .263 | .273 | .401 | .083 | .026 | .265 | .355 | .551 | .427 | .328 | .257 |
| SAR | .084 | .289 | .331 | _.373_ | .455 | .203 | .166 | .323 | .362 | .626 | .493 | **.355** | .338 |
| Semantic Density | .163 | .149 | .188 | .196 | .313 | .272 | **.357** | **.401** | .463 | .654 | .295 | .183 | .303 |
| RAUQ | .329 | **.340** | .508 | **.391** | **.554** | .331 | _.257_ | .331 | .481 | .633 | **.628** | .283 | **.422** |

*Table 24.* Detailed PRR↑ for the Falcon-3 10B model across each dataset. The best method is in **bold**, the second best is underlined. Warmer color indicates better results.

| UQ Method | XSum | SamSum | CNN | WMT14 | WMT19 | MedQUAD | TruthfulQA | CoQA | SciQ | TriviaQA | MMLU | GSM8k | Mean |
|---|---|---|---|---|---|---|---|---|---|---|---|---|---|
| MSP | .178 | .053 | **.301** | .269 | .396 | -.004 | -.001 | .300 | .459 | .674 | .621 | .364 | .301 |
| Perplexity | .141 | .152 | .248 | .355 | .524 | **.266** | .209 | .276 | .158 | .660 | .617 | .307 | .326 |
| CCP | .128 | .043 | .213 | .249 | .325 | -.041 | -.002 | .259 | .349 | .653 | .533 | .339 | .254 |
| Attention Score | **.272** | .077 | .227 | .113 | .064 | -.037 | -.024 | -.034 | -.073 | .109 | .226 | .210 | .094 |
| Focus | .159 | .069 | .187 | .262 | .463 | .123 | .208 | .218 | .304 | .656 | .486 | .195 | .278 |
| Simple Focus | .089 | .046 | .150 | .313 | .457 | .005 | .160 | .325 | .388 | **.680** | .603 | .294 | .292 |
| DegMat NLI Score entail. | .107 | .152 | .136 | .140 | .304 | .115 | .203 | .326 | .391 | .617 | .418 | **.391** | .275 |
| Ecc. NLI Score entail. | -.028 | .104 | .037 | .203 | .360 | .097 | .066 | .298 | .432 | .593 | .437 | .368 | .247 |
| EVL NLI Score entail. | .103 | **.157** | .145 | .131 | .281 | .111 | .204 | .319 | .436 | .618 | .403 | .366 | .273 |
| Lexical Similarity Rouge-L | .096 | .090 | .065 | .211 | .339 | .035 | .087 | .306 | .238 | .595 | .454 | .281 | .233 |
| EigenScore | .064 | .010 | .079 | .177 | .294 | -.067 | .104 | .283 | .336 | .542 | .357 | .173 | .196 |
| LUQ | .134 | .134 | .095 | .126 | .265 | .127 | .237 | .307 | .270 | .622 | .423 | .358 | .258 |
| Semantic Entropy | .143 | .102 | .153 | .222 | .361 | .026 | .102 | .301 | .379 | .587 | .462 | .381 | .268 |
| SAR | .084 | .119 | .079 | .256 | .419 | .070 | .180 | .308 | .253 | .650 | .514 | .364 | .275 |
| Semantic Density | .129 | .155 | .139 | .208 | .352 | .075 | **.272** | **.350** | **.524** | .620 | .352 | .314 | .291 |
| RAUQ | .151 | .156 | .235 | **.376** | **.553** | .224 | .110 | .292 | .474 | .674 | **.626** | .344 | **.351** |

# E. Additional Attention Map Examples

We provide additional examples of attention maps, similar to Figure 1, for Llama-3.1 8B and Gemma-2 9B on the QA task in Figures 5 to 8. Additionally, we provide several examples of attention maps for Llama-3.1 8B on summarization tasks from XSum (Narayan et al., 2018) in Figures 9 to 11 and translation tasks from the French-English WMT14 subset (Bojar et al., 2014) in Figures 12 to 14. These examples show that similar patterns occur consistently across multiple text instances, different models, and long-form generation tasks beyond QA.

## E.1. Question Answering

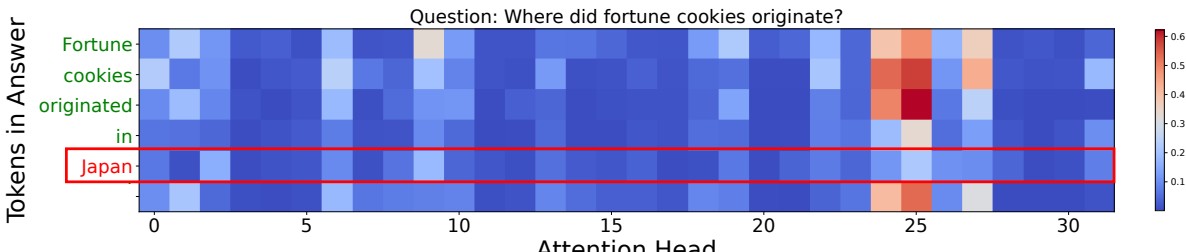

*Figure 5.* Attention weights in the 30th layer of Llama-3.1 8B from each generated token to its preceding token, given the prompt *Where did fortune cookies originate?* The $y$ axis specifies the generated tokens, and the $x$ axis specifies the attention heads. Warmer colors indicate higher attention values. The output contains the factually incorrect token *Japan* (the correct answer is either *San Francisco*, *California*, or an *unknown place*).

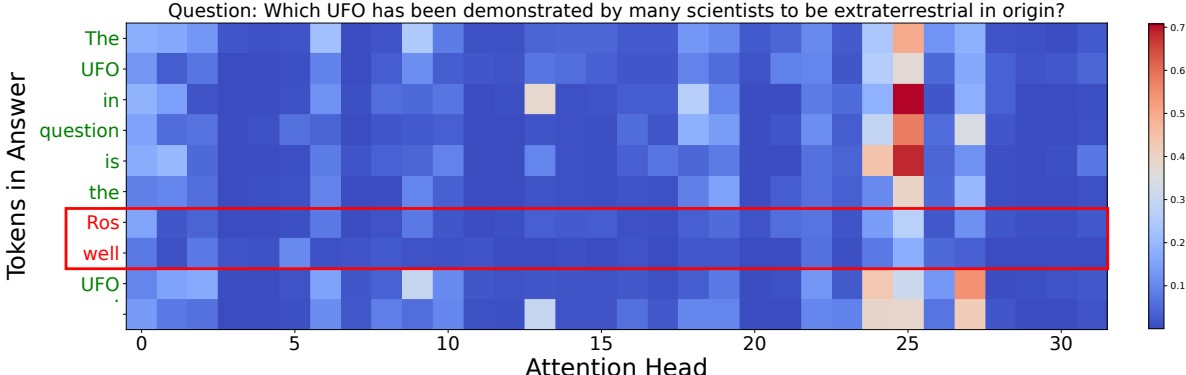

*Figure 6.* Attention weights in the 30th layer of Llama-3.1 8B from each generated token to its preceding token, given the prompt *Which UFO has been demonstrated by many scientists to be extraterrestrial in origin?*. The $y$ axis specifies the generated tokens, and the $x$ axis specifies the attention heads. Warmer colors indicate higher attention values. The output contains the factually incorrect token *Roswell UFO* (the correct answer is *There is no such UFO*).

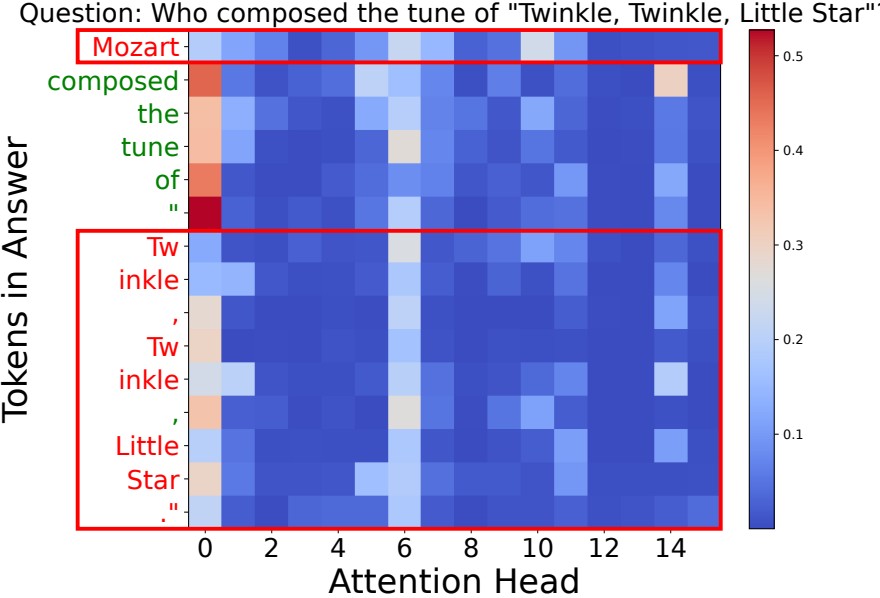

*Figure 7.* Attention weights in the 25th layer of Gemma-2 9B from each generated token to its preceding token, given the prompt *Who composed the tune of "Twinkle, Twinkle, Little Star"?*. The $y$ axis specifies the generated tokens, and the $x$ axis specifies the attention heads. Warmer colors indicate higher attention values. The output contains the factually incorrect token *Mozart* (the correct answer is either *The tune does not have a recorded composer* or *The tune of "Twinkle, Twinkle, Little Star" was a French folk song*).

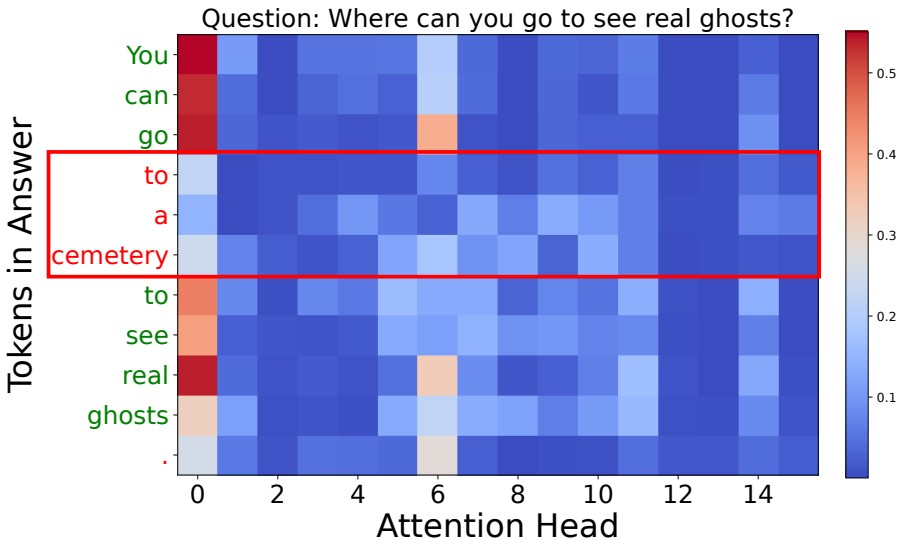

*Figure 8.* Attention weights in the 25th layer of Gemma-2 9B from each generated token to its preceding token, given the prompt *Where can you go to see real ghosts?*. The $y$ axis specifies the generated tokens, and the $x$ axis specifies the attention heads. Warmer colors indicate higher attention values. The output contains the factually incorrect token *to a cemetery* (the correct answer is *There aren't any places where you can see real ghosts*).

## E.2. Summarization

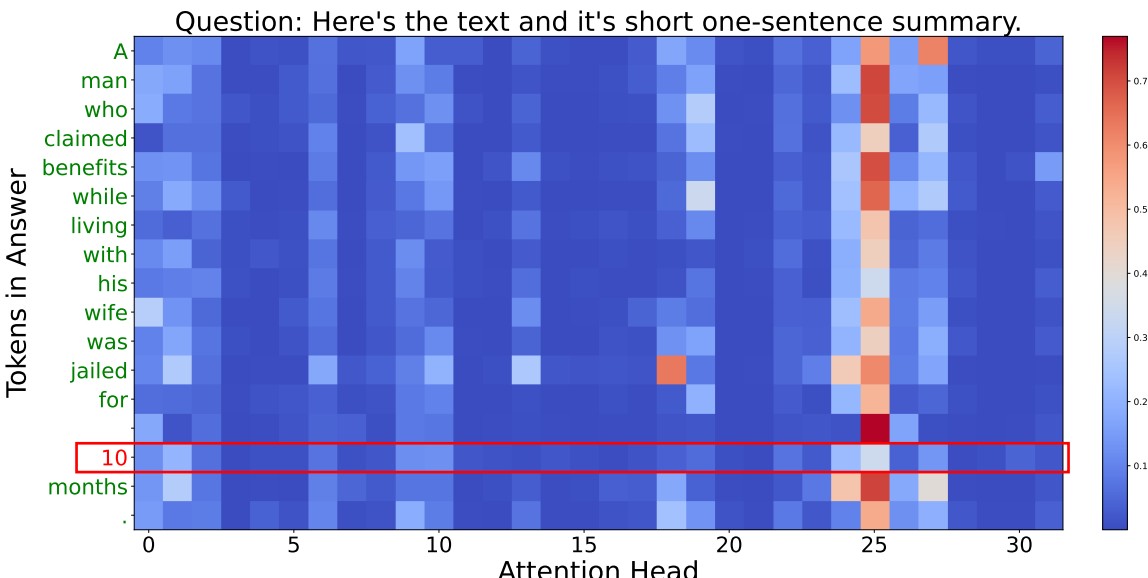

*Figure 9.* Attention weights in the 30th layer of Llama-3.1 8B from each generated token to its preceding token, given the long prompt with the original text *Here's the text and it's short one-sentence summary. Text: Alexander Reid repeatedly told Department of Work and Pensions staff in application forms and at interviews that he was single. But in reality he was living with his wife Kathleen Reid, despite having claimed to be separated. Reid was found guilty following a trial at Dundee Sheriff Court. The 59-year-old, from Dundee, had denied a charge under the Social Security Administration Act that he fraudulently claimed employment support allowance and income support totalling £39,808. Defence solicitor John Boyle asked that Reid be spared jail and given a community payback order as an alternative to a prison sentence. Sheriff Tom Hughes told Reid: "Because of the sum of money involved a custodial sentence is the only option."*. The $y$ axis specifies the generated tokens, and the $x$ axis specifies the attention heads. Warmer colors indicate higher attention values. The output contains the factually incorrect token *10* (months), while the original text does not contain information about term.

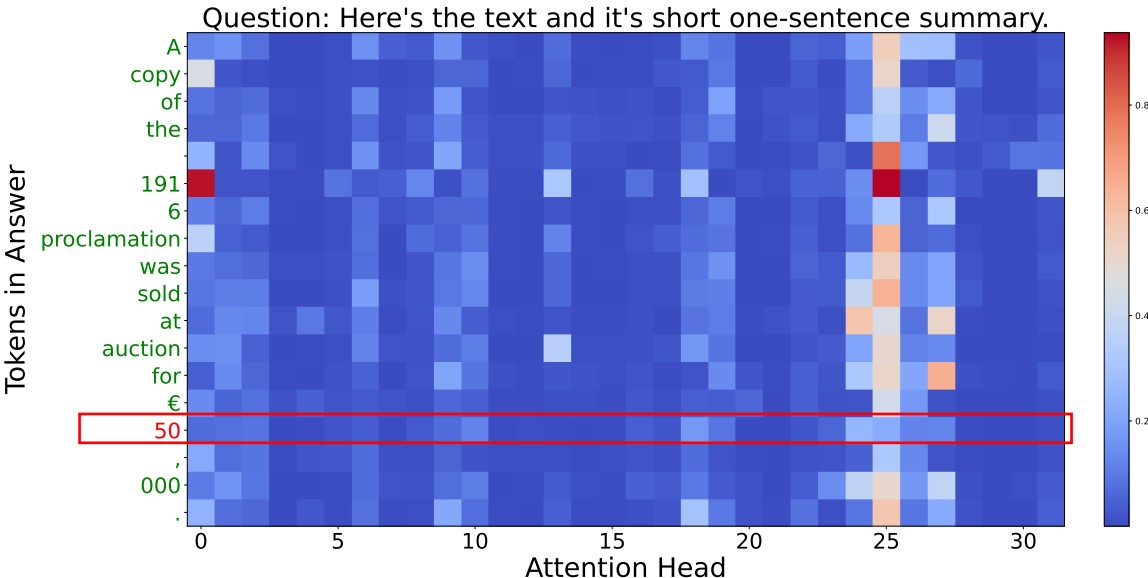

*Figure 10.* Attention weights in the 30th layer of Llama-3.1 8B from each generated token to its preceding token, given the long prompt with the original text *Here's the text and it's short one-sentence summary. Text: The item was sold to a private Irish collector. It was part of a sale of over 600 items, many from the 1916 Rising. The rebellion was an attempt to overthrow British rule 100 years ago. The 1916 proclamation is considered one of the most important documents in Irish history. Auctioneers said that the copy was from Dr James Ryan, a medical officer attached to the garrison based out of the General Post Office (GPO) during the Easter Rising. The GPO was the headquarters of the Rising's leaders. After the GPO was taken by rebel forces, Pádraig Pearse, the Rising's commander-in-chief, read the proclamation from the front of the building. The seven signatories of the proclamation were executed, along with nine other leaders, after the Rising was quelled. Last month, hundreds of thousands of people lined the streets of Dublin for a parade to mark the Easter Rising's centenary. The 1916 Proclamation was then read out by an officer from the Irish defence forces during the parade, in a re-enactment of the declaration of independence the rebels made outside the GPO.* The $y$ axis specifies the generated tokens, and the $x$ axis specifies the attention heads. Warmer colors indicate higher attention values. The output contains the factually incorrect token *50,000* (euros), while the original text does not contain information about the price of the proclamation.

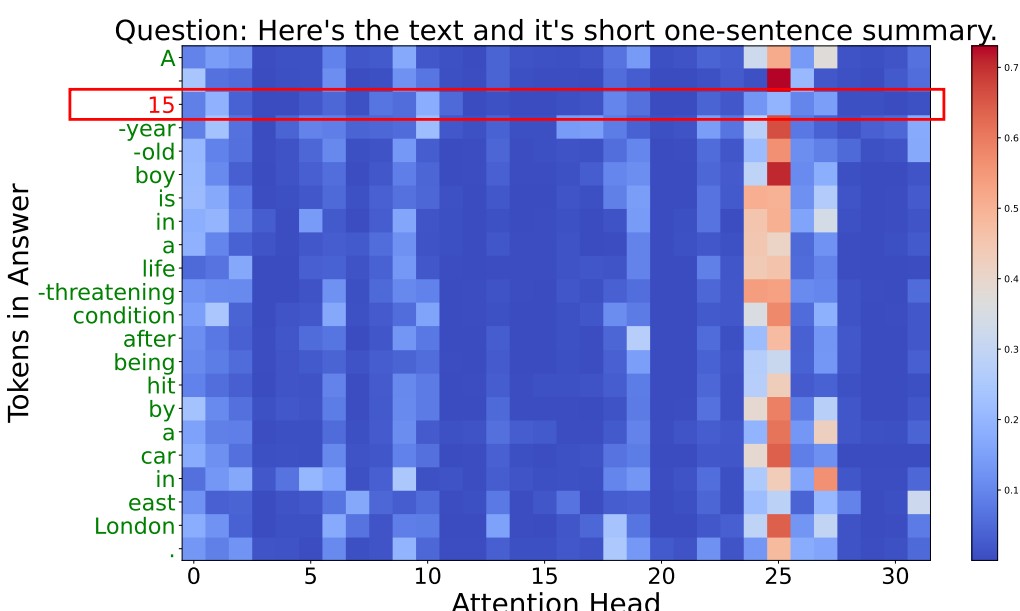

*Figure 11.* Attention weights in the 30th layer of Llama-3.1 8B from each generated token to its preceding token, given the long prompt with the original text *Here's the text and it's short one-sentence summary. Text: The victim is being treated in hospital after he was struck by the vehicle in Tower Hamlets, east London, on Friday. He was taken to a major trauma centre by London's Air Ambulance, after the crash in Bow Road, at about 08:15 GMT. A 25-year-old man was later arrested on suspicion of dangerous driving and failing to stop at the scene of an accident and bailed until February. The Met Police said on Friday evening that the boy's injuries had become life-threatening and his next of kin were aware. The car involved in the crash was later found abandoned..* The $y$ axis specifies the generated tokens, and the $x$ axis specifies the attention heads. Warmer colors indicate higher attention values. The output contains the factually incorrect token *15* (year-old boy), while the original text does not contain information about the boy's age.

## E.3. Translation

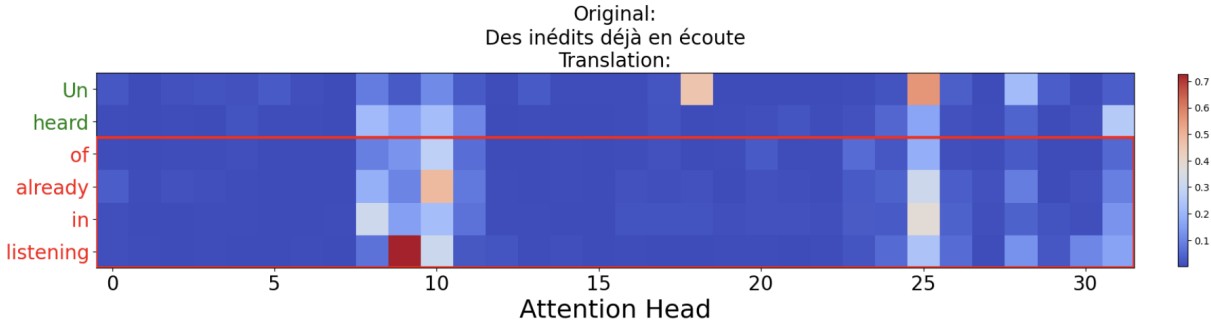

*Figure 12.* Attention weights in the 29th layer of Llama-3.1 8B from each generated token to its preceding token, given the long prompt with the original text *Here is a sentence in French language and its translation in English language. Original: Des inédits déjà en écoute Translation:*. The $y$ axis specifies the generated tokens, and the $x$ axis specifies the attention heads. Warmer colors indicate higher attention values. The output contains direct translation instead of the intended idiomatic meaning (the correct answer is *Previously Unpublished Tracks Released*).

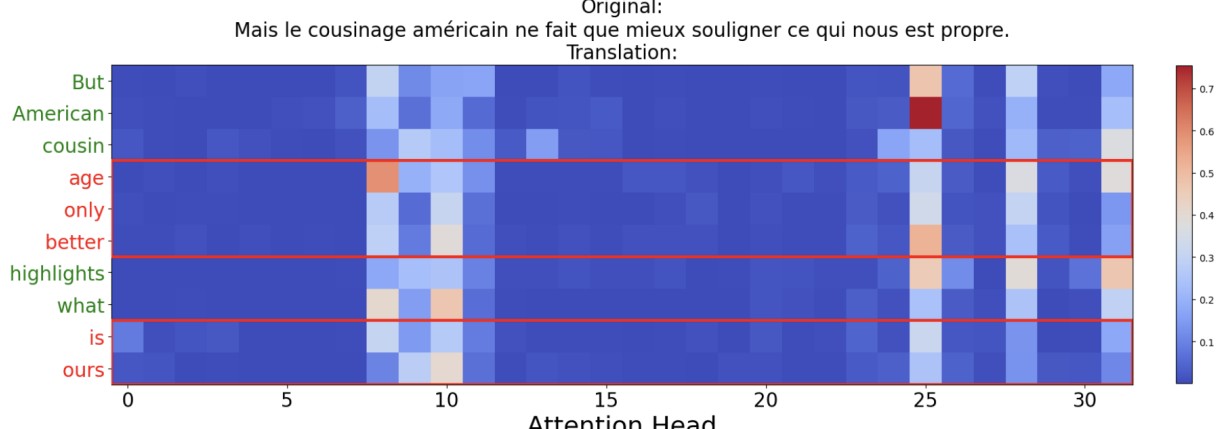

*Figure 13.* Attention weights in the 29th layer of Llama-3.1 8B from each generated token to its preceding token, given the long prompt with the original text *Here is a sentence in French language and its translation in English language. Original: Mais le cousinage américain ne fait que mieux souligner ce qui nous est propre. Translation:*. The $y$ axis specifies the generated tokens, and the $x$ axis specifies the attention heads. Warmer colors indicate higher attention values. The output contains a literal mistranslation of a culture-specific expression (the correct answer is *However, our familiarity with America only emphasises even more something that is particular to us.*).

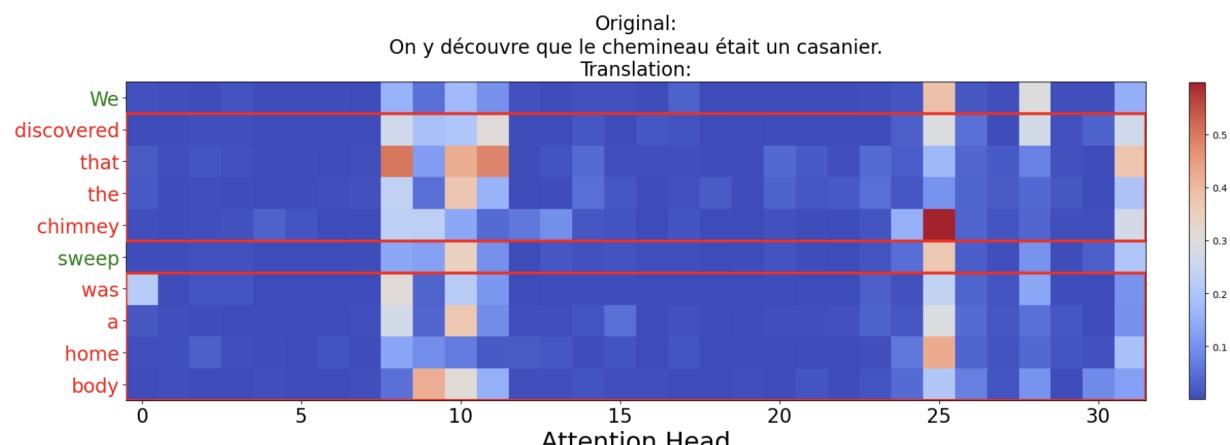

*Figure 14.* Attention weights in the 29th layer of Llama-3.1 8B from each generated token to its preceding token, given the long prompt with the original text *Here is a sentence in French language and its translation in English language. Original: On y découvre que le chemineau était un casanier. Translation:*. The $y$ axis specifies the generated tokens, and the $x$ axis specifies the attention heads. Warmer colors indicate higher attention values. The output assigns an incorrect meaning to a key source word. (the correct answer is *It reveals that the wanderer liked to stay at home.*).

## F. Error Analysis

To further investigate which generations are chosen by RAUQ, we conducted an error analysis on a small subset of the TruthfulQA dataset. To do so, we chose the top-20 samples with the highest and lowest RAUQ scores and carefully attributed the corresponding generations as truthful or erroneous. For errors, we also analyzed each error as a factual or reasoning error. The results are presented in Table 25.

*Table 25.* Error analysis for detected by RAUQ generations for Llama-3.1 8B on TruthfulQA dataset.

|  | **Erroneous generations (reasoning / factual)** | **Truthful generations** |
| --- | --- | --- |
| Samples with highest RAUQ scores | 95% (35% / 60%) | 5% |
| Samples with lowest RAUQ scores | 50% (15% / 35%) | 50% |

# G. Limitations

Our approach is unsupervised and involves only a single hyperparameter. While we demonstrate that a predefined value yields robust performance across various tasks, fine-tuning this parameter for specific datasets could lead to further improvements, which would require a validation set.

In this work, we focus on white-box UQ methods – techniques that assume full access to the internal states of an LLM. Although such methods cannot be directly applied to black-box models (e.g. LLMs exposed only through API), our work demonstrates that white-box access enables substantial performance improvements while remaining computationally efficient. Consequently, our approach paves the way for integrating robust UQ mechanisms directly into existing LLM-as-a-service systems, which is highly useful for real-world applications.

Nevertheless, one possible direction for adapting our technique to a black-box setting is to employ an auxiliary white-box proxy LLM from which attention signals and logits can be extracted. Such a proxy model may be effective because it can detect ambiguous or underspecified queries, thereby capturing uncertainty patterns that partially mirror those of the black-box target model.

