# OpenReview forum: "Efficient Hallucination Detection for LLMs Using Uncertainty-Aware Attention Heads"
_ICML.cc/2026/Conference — ICML 2026 regular_

### Official Review · Reviewer_ikdn · 2026-03-05

**Soundness:** 2
**Presentation:** 3
**Significance:** 2
**Originality:** 3
**Overall Recommendation:** 4
**Confidence:** 4

**Summary:**

This paper proposes RAUQ (Recurrent Attention-based Uncertainty Quantification), an unsupervised and efficient uncertainty quantification framework for hallucination detection in large language models (LLMs). The core observation is that within the Transformer attention mechanism, a small subset of "uncertainty-aware" attention heads tend to significantly reduce their attention to preceding tokens when the model generates hallucinated content. RAUQ automatically identifies these special attention heads and recursively fuses their activation patterns with token-level confidence scores, producing sequence-level uncertainty estimates in a single forward pass. The authors conduct systematic evaluations on four mainstream open-weight LLMs (Llama-3.1 8B, Qwen-2.5 7B, Gemma-2 9B, Falcon-3 10B) and 12 benchmark datasets spanning question answering, summarization, and translation. Results show that RAUQ consistently outperforms 15 baseline methods on the PRR metric, requiring neither labeled data nor task-specific hyperparameter tuning, and exhibits strong plug-and-play characteristics.

**Compliance With Llm Reviewing Policy:**

Affirmed.

**Key Questions For Authors:**

1. This paper is entirely based on empirical observations in proposing to use only the attention weight to the last preceding token for hallucination detection. Could the authors provide a more direct mechanistic explanation for why the attention weight to the preceding token ($i-1$) is better able to reflect the model's factual uncertainty?

2. Figure 3(c) only shows the attention score differences for the preceding 6 tokens, where the difference at position $i-1$ is the largest. Could the authors supplement this with the attention score differences from token $i$ to itself, or to all tokens in the sequence, in order to more convincingly support the motivation for selecting only the immediately preceding token?

3. I notice that RAUQ does not appear to hold a clear advantage over PPL in terms of ROC-AUC. Given that most hallucination detection papers report ROC-AUC as the primary metric, could the authors provide more detailed results, such as a comprehensive comparison with PPL and other methods across all 12 datasets? Additionally, since the paper only considers base models, could the authors supplement the results with experiments on instruction-tuned models such as Llama-3.1-Instruct, evaluated under the ROC-AUC metric?

**Limitations:**

yes

**Strengths And Weaknesses:**

### Strengths

1. RAUQ requires only a single forward pass, without relying on multiple sampling rounds or auxiliary models, introducing less than 1% additional computational overhead (Table 4). Compared to sampling-based methods such as Semantic Entropy and SAR that incur 400%–800% extra overhead, RAUQ demonstrates clear practical advantages in latency-sensitive production environments.

2. The experiments cover 4 mainstream LLMs and 12 cross-task datasets (QA/summarization/translation), with additional results in the appendix ranging from 360M to 70B model sizes, thoroughly validating the method's generalization across models and tasks. The ablation studies systematically analyze individual components (attention head selection, recurrent propagation, layer aggregation strategy, and hyperparameter sensitivity), providing a reasonably complete justification.

3. In contrast to supervised methods such as SAT Probe, Lookback Lens, and TAD that require task-specific labeled training data, RAUQ is fully unsupervised and significantly outperforms supervised baselines in out-of-domain scenarios (Tables 14 and 15), demonstrating stronger practical deployment flexibility.

---

### Weaknesses

1. The primary weakness of this paper is the insufficient mechanistic motivation. The core design of RAUQ—using only the attention weight to the immediately preceding token ($i-1$) as a hallucination signal—is entirely based on empirical observation, lacking more direct mechanistic or theoretical justification. The authors cite the eigenvalue analysis from Sriramanan et al. (2024) as supporting evidence; however, that work uses attention weights of each token to **itself** ($i$), not to the previous token ($i-1$), because these weights correspond to the eigenvalues of the lower-triangular attention matrix and their sum equals exactly the log-determinant of that matrix. The authors appear to have misunderstood this point.

2. Figure 3(c) presents the difference in attention weights to the preceding 6 tokens ($i-6$ to $i-1$) and uses this to argue for the superiority of the $i-1$ position. However, this difference may not stem from better discriminability of hallucination patterns, but simply from the fact that attention weights at position $i-1$ are inherently much larger in magnitude than those at earlier positions, naturally leading to larger absolute differences.

3. This paper adopts PRR (Prediction Rejection Ratio) as the primary evaluation metric, whereas most papers in the hallucination detection literature report ROC-AUC as the core metric. From Appendix Table 13, RAUQ's advantage over simple baselines such as Perplexity in terms of ROC-AUC is not particularly pronounced, raising concerns about the method's actual discriminative capability.

4. Furthermore, the experiments are conducted exclusively on base models, without evaluation on instruction-tuned models (e.g., Llama-3.1-Instruct), whereas instruction-tuned models are far more commonly used in real-world deployment scenarios.

---

> ### Author Rebuttal · Authors · 2026-03-31
>
> Thank you very much for your time in reviewing our paper and your very valuable comments.
>
> **Q1: The work of Sriramanan et al. 2024 uses attention weights of each token to itself, not to the previous token**
>
> We want to ensure that our interpretation is fully consistent with Sriramanan et al., the difference lies only in notation.
> In our notation, token $i-1$ has already been generated, and the model is generating token $i$, for which we assess the presence of hallucination. We refer to attention $i-1$ as **attention to the previous token**, because the signal is used to detect hallucination in the token being generated.
> In the notation of Sriramanan et al., token $i$ has already been generated, and the model is generating token $i+1$, which they describe as **attention to itself**. Although this usage appears in the literature, the signal in fact reflects the likelihood of hallucination for token $i+1$, not for the token $i$ “itself.”
> This shifts the attention matrix indexing, but all underlying backgrounds remain unchanged. We also show that while Attention Score performs well, head selection via our method (Table 2) improves it further. We will clarify this in the camera-ready version.
>
> **Q2. Figure 3(c): … attention weights at position $i-1$ are inherently much larger in magnitude. …supplement this attention differences from token $i$ to itself, or to all tokens in the sequence**
>
> Figure 3(a) shows both absolute and relative differences, confirming the first token's difference is substantially larger even in percentage terms.
>
> Token $i$ that is being generated has no attention to “itself”, we measure its attention to $i-1$ during the generation of token $i$, as denoted in the figure.
>
> Only six preceding tokens are shown for clarity. Showing more is impractical as TruthfulQA generations vary up to 128 tokens.
>
> Relative percentages for Figure 3(c) from $i-6$ to $i-1$ are 1.3\%, 3.2\%, 6.1\%, 3.7\%, 8.4\%, 11.7\% further confirming the effect is not merely due to larger absolute weights at later positions. We will include these values in the camera-ready version.
>
> **Q3. This paper adopts PRR, while most papers in the literature report ROC-AUC. Detailed comparison … across all 12 datasets**
>
> PRR metric is a part of the common practice for benchmarking UQ methods (Malinin and Gales, Vashurin et al. 2025, Santilli et al. 2025, Vazhentsev et al. 2025, Bakman et al. 2025, Song et al. 2025), as discussed starting from line 328.
>
> ROC-AUC is not a good metric for benchmarking UQ methods for the following reasons:
> (1) ROC-AUC is unreliable for highly imbalanced tasks such as hallucination detection. (2) ROC-AUC requires binary targets, so it requires to develop thresholds for continuous quality metrics which are common in MT, summarization, and other tasks. (3) It treats the rejection of 10\% and 90\% of generations equally. Apparently, rejection of 90% is not a very good use case as noted in line 279. PRR addresses this by covering only the first 50\% of the rejection curve.
>
> Therefore, PRR remains more reliable, while differences in ROC-AUC could be caused by artifacts of this metric.
>
> Even, according to ROC-AUC our method outperforms many baselines. Table 13 shows that all sampling-based methods fall well behind perplexity, while RAUQ still achieves the best average results with almost no overhead.
> Full results comparing PPL and other baselines across all 12 datasets are in Appendix E, Tables 17-20.
>
> **Q4. Regarding the experiments on instruction-tuned models**
>
> We present below a table comparing RAUQ with other sampling-free baselines from Table 1, specifically for LLaMa-3.1-8B-Instruct.
> | Methods | QA | ATS | NMT | Mean |
> |-|-|-|-|-|
> | MSP | 0.358 | 0.142 | 0.336 | 0.279 |
> | Perplexity | 0.345 | 0.153 | 0.459 | 0.319 |
> | Attention Score | 0.081 | **0.201** | 0.085 | 0.122 |
> | Focus | 0.340 | 0.115 | 0.391 | 0.282 |
> | Simple Focus | 0.372 | 0.108 | 0.385 | 0.288 |
> | RAUQ | **0.372** | 0.152 | **0.471** | **0.332** |
> RAUQ outperforms other methods on QA and NMT, on ATS it matches perplexity. The results confirm that this attention pattern generalizes across instruction-tuned models and is not model- or task-specific. We will add this experiment in the camera-ready version.
>
> **Q5. Direct mechanistic explanation for why the attention weight to the preceding token is better able to reflect the model's factual uncertainty?**
>
> The theoretical explanation for why attention to the preceding token better reflects factual uncertainty is already presented in Section 4 (lines 202–261).
>
> The **relevant signal** is the drop of attention to $t-1$ relative to the typical attention to $t-1$ for the given head; finding this “change point” can be implemented in various ways.
>
> Our implementation briefly: if the attention from the current token $t$ to $t-1$ is lower than the average attention to $t-1$ across all time steps, $t$ is a hallucination. We look only at indicative heads with the highest average attention to $t-1$.

---

> > ### Author Rebuttal · Reviewer_ikdn · 2026-04-01
> >
> > Thank you for the response, which clarified some issues. I have updated my scores accordingly.

---

> > > ### Author Response · Authors · 2026-04-08
> > >
> > > Thank you very much for your thoughtful feedback and for updating the score! We are glad that our rebuttal fully addressed your concerns, and we will make sure to incorporate all these revisions and results in the paper.

---

### Official Review · Reviewer_yAar · 2026-03-11

**Soundness:** 3
**Presentation:** 3
**Significance:** 3
**Originality:** 3
**Overall Recommendation:** 4
**Confidence:** 3

**Summary:**

The paper introduces RAUQ, an unsupervised framework designed to detect factual inaccuracies in LLMs using attention. The paper hinges on a specific observation about transformer attention mechanisms: when an LLM generates incorrect information, certain "uncertainty-aware" attention heads within the model tend to reduce their focus on preceding tokens. RAUQ capitalizes on this by automatically identifying these specific attention heads. It then combines their activation patterns with standard token-level confidence measures using a recurrent scheme. This allows the framework to generate a sequence-level uncertainty estimate in just a single forward pass. Across twelve distinct tasks and using four different LLMs, the authors demonstrate that it outperforms state-of-the-art uncertainty quantification (UQ) baselines.

**Compliance With Llm Reviewing Policy:**

Affirmed.

**Final Justification:**

The authors have clarified my concerns. I think putting these clarifications into the paper would make it a stronger paper.

**Key Questions For Authors:**

1. How reliably can RAUQ identify "uncertainty-aware" attention heads in highly complex or specialized architectures, such as Mixture-of-Experts (MoE) models?
2. Just curious, how would it work for estimating black-box adaptation? Why is there a guarantee that a model can simulate the attention mechanisum of another model, especially considering that powerful proprietary models are complex?
3. Does RAUQ perform equally well on all types of hallucinations? For instance, can it detect logically flawed reasoning where the model remains internally confident, or does it primarily excel at catching factual recall failures?

**Limitations:**

yes

**Strengths And Weaknesses:**

## Strengths

- **Novel Use of Attention Mechanics:** Leveraging the internal behavior of "uncertainty-aware" attention heads is an innovative approach.
- **Good Computational Efficiency**: One advantage of RAUQ is its speed. By extracting uncertainty estimates in a single forward pass, it incurs less than 1% additional computational overhead.
- **Strong Empirical Performance**: The methodology is rigorously tested across a diverse set of twelve tasks and four different model architectures, proving consistent improvements over baselines.

## Weaknesses

- **Hallucination Evaluation**: The paper claims that it targets hallucinations but for PRR other quality metrics that do not measure hallucinations are used, such as accuracy, and COMET.
- **Small sample size of analysis**: The analysis to find the trend seems to only stem from observations on 20 examples which is small and only using a single model. A more comprehensive quantitative analysis to show that the attention showcases the specific hallucination trend would be helpful.
- **Generalizability to other architectures**: Given how many of the current models now work as a mixture of expert models, showing analysis for such models would also be interesting and providing more useful takeaways.
- **Robustness of model size**: The experiments have been conducted on different models of similar number of parameters. It might be interesting to show whether this attention mechanism still holds for a larger or smaller model.

---

> ### Author Rebuttal · Authors · 2026-03-31
>
> Thank you very much for your time in reviewing our paper and your very valuable comments.
>
> **Q1. The experiments have been conducted on different models of similar number of parameters.**
>
> Here, we respectfully disagree. We experiment with LLMs of various sizes: 360M (Table 12), 1B (Table 12), 7-10B (Table 1), 70B (Table 12), and a new experiment with 21B (see the following answer).
>
> RAUQ delivers the best performance across models of various sizes.
>
> **Q2. How reliably can RAUQ identify "uncertainty-aware" attention heads in highly complex or specialized architectures, such as Mixture-of-Experts (MoE) models?**
>
> (1) MoE does not change the pattern, because its roots lie in the training objective and general principles of conditional generation in LLMs.
>
> We conduct an additional experiment in a table below, where we compare RAUQ with strong baselines from Table 1, for GPT-OSS-20B -- a 21B MoE model.
>
> | Method | GSM8k | MMLU | SamSum | Mean |
> |-|-|-|-|-|
> | MSP | 0.593 | **0.715** | **0.328** | *0.545* |
> | Perplexity | 0.426 | **0.715** | 0.282 | 0.474 |
> | Focus | -0.082 | 0.064 | -0.034 | -0.017 |
> | Attention Score | 0.598 | 0.046 | 0.275 | 0.307 |
> | EVL NLI Score entail. | 0.164 | 0.631 | 0.085 | 0.293 |
> | DegMat NLI Score entail. | 0.217 | 0.635 | 0.088 | 0.313 |
> | Semantic Entropy | 0.494 | 0.597 | 0.251 | 0.447 |
> | SAR | 0.389 | 0.625 | 0.253 | 0.422 |
> | Semantic Density | 0.095 | 0.153 | 0.278 | 0.175 |
> | RAUQ | **0.640** | *0.711* | *0.312* | **0.554** |
>
> RAUQ ranks first across all tasks, which suggests that a similar hallucination pattern holds for MoE models.
>
> (2) We also show that RAUQ does not depend specifically on attention: it can be replaced by interpretability signals such as Layer Integrated Gradients with comparable performance (see Table 16 in Appendix D4). This suggests that RAUQ is not tied to a particular attention implementation and may be applicable to architectures without attention.
>
> **Q3. The analysis to find the trend seems to only stem from observations on 20 examples which is small …**
>
> The analysis in Figure 2 considers only 20 observations for illustrative purposes.
>
> Figures 3(a) - 3(c) provide a more detailed quantitative analysis, including additional instances from TruthfulQA. These figures reveal clear attention patterns across all examples.
>
> Moreover, in Appendix F (Figures 7 and 8), we also present attention maps for Gemma-2 9, which exhibit the same patterns as observed for LLaMa.
>
> **Q4. Does RAUQ perform equally well on all types of hallucinations? can it detect logically flawed reasoning?**
>
> Our experiments demonstrate robustness across 12 datasets across 3 task types, **including reasoning** (see Appendix A.2, lines 731–762, for a detailed task description).
>
> RAUQ performance specifically for CoT reasoning is verified on GSM8K (Tables 17-20).
>
> Moreover, we validate its performance on adversarial inputs of TruthfulQA in Tables 17-20 in Appendix E.
>
> Therefore, RAUQ remains highly competitive across a diverse range of hallucination types.
>
> **Q5. … quality metrics that do not measure hallucinations are used, such as accuracy, and COMET.**
>
> We follow the common practice for benchmarking UQ methods, see e.g.: (Malinin and Gales 2021, Vashurin et al. 2025, Santilli et al. 2025, Bakman et al. 2025, Song et al. 2025). Generation quality metrics: accuracy, COMET, etc. serve as a proxy for sequence-level hallucinations.
>
> **Q6. how would it work for estimating black-box adaptation?**
>
> One possible direction for adapting RAUQ to a black-box setting is to use a white-box proxy LLM from which attention signals and logits can be extracted. We discuss this in Appendix H.
>
> **Q7. Why is there a guarantee that a model can simulate the attention mechanisum of another model…?**
>
> White-box proxy can approximate the aleatoric uncertainty related to ambiguity and complexity of the question.
>
> Zhang et al., 2023 and Sriramanan et al., 2024 have already tested this approach. Given that RAUQ substantially outperforms these methods, we can expect that it can be successfully applied in a black-box setting as well.
>
> **References:**
>
> [1] Santilli et al. Revisiting Uncertainty Quantification Evaluation in Language Models: Spurious Interactions with Response Length Bias Results. ACL 2025. \
> [2] Sriramanan et al. LLM-Check: Investigating Detection of Hallucinations in Large Language Models. NeurIPS 2024.  \
> [3] Vashurin et al. Benchmarking Uncertainty Quantification Methods for Large Language Models with LM-Polygraph. TACL 2025. \
> [4] Zhang et al. Enhancing Uncertainty-Based Hallucination Detection with Stronger Focus. EMNLP 2023.
> [5] Bakman et al. "Reconsidering LLM uncertainty estimation methods in the wild." ACL 2025. \
> [6] Song et al. "Inv-Entropy: A Fully Probabilistic Framework for Uncertainty Quantification in Language Models." NeurIPS 2025. \
> [7] Malinin and Gales. "Uncertainty Estimation in Autoregressive Structured Prediction." ICLR 2021.

---

> > ### Author Rebuttal · Reviewer_yAar · 2026-04-01
> >
> > Thank you for the detailed response. Incorporating these clarifications would make the paper stronger. I have updated my scores accordingly.

---

> > > ### Author Response · Authors · 2026-04-08
> > >
> > > Thank you very much for your thoughtful feedback and for updating the score! We are glad that our rebuttal fully addressed your concerns, and we will make sure to incorporate all these clarifications in the paper.

---

### Official Review · Reviewer_qy6s · 2026-03-13

**Soundness:** 3
**Presentation:** 3
**Significance:** 3
**Originality:** 3
**Overall Recommendation:** 4
**Confidence:** 2

**Summary:**

This paper proposes Recurrent Attention-based Uncertainty Quantification (RAUQ), an efficient, unsupervised method for detecting hallucinations in LLMs. The authors identify that specific "uncertainty-aware" attention heads systematically reduce their attention to the preceding token when generating factually incorrect information. Based on this mechanistic observation, RAUQ dynamically selects these informative heads and recurrently fuses their attention weights with token-level probabilities to estimate sequence-level uncertainty in a single forward pass. Extensive experiments across 12 datasets and multiple LLMs demonstrate that RAUQ achieves state-of-the-art uncertainty quantification performance over 15 baselines. Importantly, it operates entirely without labeled data or multiple sampling passes, adding less than 1% computational overhead to standard inference.

**Compliance With Llm Reviewing Policy:**

Affirmed.

**Final Justification:**

Most of my concerns have been addressed. Therefore, I maintain the score.

**Key Questions For Authors:**

1. Could the authors provide attention map visualizations and comprehensive error analyses for complex, long-form tasks such as abstractive summarization and machine translation ? It is essential to understand how "uncertainty-aware" heads behave when sequence-level errors are distributed across multiple tokens rather than localized to a single factual entity.

2. While Appendix D.1 provides PRR performance metrics for the LLaMA-3.1 70B model, the mechanistic behavior inside these larger models remains unexplored. Does the core observation—that specific attention heads sharply reduce focus on preceding tokens during hallucinations—hold consistently for models of this scale ? Do the attention patterns and the concentration of signal within a single head  mirror those observed in the 8B models?

**Limitations:**

yes

**Strengths And Weaknesses:**

**Strengths**
1. The paper is clearly written with a logical narrative flow that connects empirical observations directly to the algorithm design.
2. The core observation that specific "uncertainty-aware" attention heads systematically reduce focus on preceding tokens during hallucinations provides a valuable mechanistic insight.
3. The experimental evaluation is exceptionally thorough, testing the proposed RAUQ method across twelve distinct datasets and four different large language models.

**Weaknesses**
1. The qualitative evaluation lacks comprehensive, visualized case analyses for complex, long-form generation tasks such as abstractive summarization and machine translation .
2. The current error analysis is limited to a small, isolated subset of samples exclusively from the TruthfulQA dataset . Expanding this qualitative analysis to other generation tasks, such as summarization or translation, would provide a deeper understanding of the algorithm's broader failure modes.

---

> ### Author Rebuttal · Authors · 2026-03-31
>
> Thank you very much for your time in reviewing our paper and your insightful comments.
>
> **Q1: Could the authors provide attention map visualizations and comprehensive error analyses for complex, long-form tasks such as abstractive summarization and machine translation ?**
>
> Visualization is given for QA for simplicity. Experimental results show effectiveness of the approach for QA and for various other tasks.
>
> Long-form tasks such as summarization and translation present a different challenge: hallucinations in these settings are typically more diffuse, often arising from phrases that are difficult to translate across languages or contexts, rather than from a single incorrect token. This makes direct visualization less interpretable.
>
> However to address your concern and enhance the motivation, we will include examples of text segments from long-form generations in the camera-ready version of the paper.
>
> **Q2: While Appendix D.1 provides PRR performance metrics for the LLaMA-3.1 70B model, the mechanistic behavior inside these larger models remains unexplored. Does the core observation—that specific attention heads sharply reduce focus on preceding tokens during hallucinations—hold consistently for models of this scale ? Do the attention patterns and the concentration of signal within a single head mirror those observed in the 8B models?**
>
> We provide attention pattern analyses for multiple LLM families, and the hallucination patterns remain consistent across them.
>
> This suggests that the behavior is likely intrinsic to the training objective and the general principles of attention mechanism rather than an artifact of a particular attention implementation.
>
> Our illustrations for LLaMA-3.1 8B are shown in Figures 1-3 and 5-6 in Appendix F, while those for Gemma-2 9B appear in Figures 7-8 in Appendix F.
>
> We note that generating full attention-head visualizations for the 70B model is not practical for the paper given the large number of layers and heads involved, but the experimental results confirm that the attention pattern generalizes to this scale as well.
>
> Nevertheless, we will provide the attention maps for selected heads for bigger LLMs in the camera ready version.
>
> **Q3. The current error analysis is limited to a small, isolated subset of samples exclusively from the TruthfulQA dataset . Expanding this qualitative analysis to other generation tasks, such as summarization or translation, would provide a deeper understanding of the algorithm's broader failure modes.**
>
> TruthfulQA is used for error analysis because its medium-length, and factual answers allow precise manual labeling of errors as factual or reasoning mistakes.
>
> In summarization and translation, hallucinations are more diffuse, often arising from complicated contexts rather than from a single factually incorrect token, making reliable manual annotation more challenging.
>
> According to your suggestion, we will extend the error analysis by including visualizations of attention patterns on representative text segments from summarization and translation tasks in the camera-ready version.

---

> > ### Author Rebuttal · Reviewer_qy6s · 2026-04-02
> >
> > Thank you for the response. Most of my concerns have been addressed. I hope the promised revisions will be incorporated into the final version of the paper.

---

> > > ### Author Response · Authors · 2026-04-08
> > >
> > > Thank you very much for your positive evaluation of our paper! We are glad that our rebuttal fully resolved your concerns and we will make sure to include all these promised revisions in the paper.

---

### Official Review · Reviewer_UC6Q · 2026-03-13

**Soundness:** 3
**Presentation:** 3
**Significance:** 3
**Originality:** 2
**Overall Recommendation:** 5
**Confidence:** 2

**Summary:**

The paper proposes Recurrent Attention-based Uncertainty Quantification (RAUQ), an unsupervised and efficient method for detecting hallucinations in large language models. The approach is based on the observation that when models generate incorrect information, certain attention heads reduce their focus on preceding tokens. RAUQ automatically identifies these uncertainty-aware heads and combines their activation patterns with token-level confidence scores in a recurrent framework to produce a sequence-level uncertainty estimate using a single forward pass. Experiments across twelve tasks, including question answering, summarization, and translation, and four different LLMs show that RAUQ outperforms existing uncertainty quantification baselines while requiring less than 1% additional computation. Because the method does not rely on labeled data or extensive parameter tuning, it provides a lightweight and practical approach for real-time hallucination detection in white-box LLMs.

**Compliance With Llm Reviewing Policy:**

Affirmed.

**Key Questions For Authors:**

See Weaknesses.

**Limitations:**

Yes.

**Strengths And Weaknesses:**

### Strengths
While the main insight appears largely empirical, the proposed approach is intuitive as a **white-box method for uncertainty quantification**. The authors clearly articulate the key observation that motivates the approach and provide a logical progression leading to the construction of **Algorithm 1**, which helps make the method easy to follow.

### Weaknesses
At times, the figures and the narrative appear somewhat **inconsistent or confusing**. For example, the authors suggest in **Figure 3c** that in practice it is often sufficient to look only one or two tokens back. If this is the case, it raises the question of why the authors did not adopt a **change-point or delta-based detection method**, which would seem consistent with the narrative description of the phenomenon. It would be helpful if the authors could clarify the reasoning behind the chosen approach.

Additionally, the authors state:

> "For most attention heads, the weights to previous tokens remain low across all generated tokens. In contrast, the 25th head exhibits a distinct pattern: it assigns relatively high attention to the preceding token for non-hallucinated (i.e., correct) tokens, but this attention drops substantially for the hallucinated token *falcon*."

This pattern is clearly illustrated in **Figure 1**. However, **Figure 2** seems to suggest that **attention head 28** may display a stronger signal than **head 19**, which raises some confusion about the head selection. Could the authors provide the **exact average attention values for each head** for the examples shown? Reporting these values may help clarify why the specific heads highlighted in the paper were selected.

---

> ### Author Rebuttal · Authors · 2026-03-31
>
> Thank you very much for your time in reviewing our paper and your insightful comments.
>
> **Q1. Figure 3c [shows] it is often sufficient to look only one or two tokens back. … it raises the question of why the authors did not adopt a change-point or delta-based detection method, which would seem consistent with the narrative description of the phenomenon.**
>
> (1) we **do not** claim that the difference in attention to $t-1$ and $t-2$ is itself indicative of hallucination.
>
> (2) The **relevant signal** is the drop of attention to $t-1$ relative to the typical attention to $t-1$ for the given head; finding this “change point” can be implemented in various ways.
>
> Our implementation briefly: if the attention from the current token $t$ to $t-1$ is lower than the average attention to $t-1$ across all time steps, $t$ is a hallucination and if a subsequent token strongly attends to an already hallucinated token, it might be a continuation of the hallucinated span.We look only at indicative heads with the highest average attention to $t-1$.
>
> This method is fast, simple, and plug-and-play, making it applicable to any new task or model. It does not need any additional change-point or delta-based detection. We already ablated several implementations of this idea in Appendix C1, which all work worse than the suggested approach.
>
> Nevertheless, we will provide an additional experiment with a delta between attention to $t-1$ and a sliding average of attentions to $t-1$.
>
> **Q2. However, Figure 2 seems to suggest that attention head 28 may display a stronger signal than head 19, which raises some confusion about the head selection. Could the authors provide the exact average attention values for each head for the examples shown? Reporting these values may help clarify why the specific heads highlighted in the paper were selected.**
>
> Thank you for pointing this out! Our method effectively selects the 28th attention head for the 23rd layer, while highlighting head 19 appears to be a visualization issue.
>
> Here is the exact average attention values for 19th and 28th head from the Figure 2:
>
> | | Head 19 | Head 28 |
> | -- | -- | -- |
> | Correct | 0.05 | **0.20** |
> | Correct | 0.04 | **0.18** |
> | Correct | 0.07 | **0.10** |
> | Correct | 0.05 | **0.20** |
> | Correct | 0.04 | **0.16** |
> | Correct | 0.03 | **0.15** |
> | Correct | 0.05 | **0.17** |
> | Correct | 0.03 | **0.13** |
> | Correct | 0.02 | **0.11** |
> | Correct | 0.05 | **0.20** |
> | **Average** | 0.04 | 0.17 |
>
> | | Head 19 | Head 28 |
> | -- | -- | -- |
> | Incorrect | 0.03 | **0.10** |
> | Incorrect | 0.03 | **0.15** |
> | Incorrect | 0.02 | **0.08** |
> | Incorrect | 0.02 | **0.04** |
> | Incorrect | 0.02 | **0.04** |
> | Incorrect | 0.02 | **0.10** |
> | Incorrect | 0.03 | **0.11** |
> | Incorrect | 0.01 | **0.06** |
> | Incorrect | 0.02 | **0.10** |
> | Incorrect | 0.01 | **0.04** |
> | **Average** | 0.02 | 0.09 |
>
> We will fix the figure in the camera-ready version of the paper.

---

> > ### Author Rebuttal · Reviewer_UC6Q · 2026-04-02
> >
> > Thank you for implementing the proposed changes. I keep my score.

---

> > > ### Author Response · Authors · 2026-04-08
> > >
> > > Thank you very much for your positive evaluation of our paper! We are glad that our rebuttal fully resolved your concerns and we will make sure to include all these changes in the paper.

---

### Decision · Program_Chairs · 2026-04-30

**Decision:**

Accept (regular)

**Comment:**

The paper proposes an unsupervised method for detecting hallucinations for white-box LLMs. Most reviewers' concerns have been addressed during rebuttal.

- The detection method is efficient using a single forward pass and requiring less than 1% additional computation.

- Evaluation is comprehensive across twelve tasks and four different LLMs.

- The analysis could be further extended to other generation tasks to provide more solid evidence.

- Since this work comes from the empirical observations, it would be better to justify the connection between the motivation and observation.